# Simulated microgravity confines and fragments the straw-based lignocellulose degrading microbial community

Boyang Liao,[1,2] Tianyi Feng,[1,2] Sihan Hou,[1,2] Hong Liu,[1,2,3] Jiajie Feng[1,2,3]

**ABSTRACT** Crewed long-term and long-distance missions are the undoubted trends of human space exploration, which require a bioregenerative life support system (BLSS) and its efficient treatment of the highly lignocellulosic organic solid waste under microgravity. Under normal gravity and simulated microgravity effect (mμ-g) created by clinostats, we used the inoculum from the world's longest BLSS experiment "Lunar Palace 365" to ferment and degrade wheat straw. The straw and its lignocellulose contents' weight losses were significantly slowed down by mμ-g. By high-throughput sequencing and metabolomics on the fermentation material, we found that mμ-g largely shrank and fragmented the microbial community's phylogenetic molecular ecological network (pMEN), and enriched many reported antimicrobial metabolites, especially against fungi, the principal lignocellulose degrader (e.g., cyclohexylamine, an antifungal chemical, increased by 188 times). Inspired by the solid-media visualization experiment of *Aspergillus nidulans* (a representative fungus) which showed a confined hyphal expansion under mμ-g, we proposed a material-convection-based model: the degradation of complex and recalcitrant macromolecules like lignocellulose is a multistep and highly coordinative task for the microbial community, but the mμ-g physically destroyed the material convection in the fermentation material, which confined the diffusion of microbial cells, their metabolic products/substrates, and extracellular enzymes, thus fragmenting the microbial interactions needed for the degradation; the confined diffusion also caused a local resource shortage for next-step degraders, which resulted in a zonal concentration of microbes and thus intensified conflicts manifested in the release of antimicrobial metabolites, especially against fungi.

**IMPORTANCE** This convection-based model explains the observed phenomena and suggests proper mass-transfer-promoting methods for more "globalized" microbial interactions in such a community-based, highly coordinative, oligotrophic, mixed-phase (physically), and fungi-dominant application scenario under microgravity. The higher lignocellulose degradation efficiency thus achieved would certainly improve the bioregenerative life support system (BLSS) required for long-term space exploration missions. For non-space-exploration scenarios, this model could also serve as an additional illustration of both the biological and physical principles of such multistep bioprocesses.

**KEYWORDS** crewed space exploration, bioregenerative life support systems, organic solid waste, material convection, microbial community, clinostat, metabolomics

Space exploration is an important cause related to the future of mankind, and according to historical experience, it is also a considerable driving force on the Earth's economy and scientific research. Crewed long-distance missions are the inevitable development trend of human space exploration, which requires the transport or *in situ* regeneration of a large amount of life support material. All the supplies being

Address correspondence to Jiajie Feng, fengjiajie@buaa.edu.cn, or Hong Liu, lh64@buaa.edu.cn.

Boyang Liao and Tianyi Feng contributed equally to this article. The author responsible for the final processing of the manuscript is placed first.

The authors declare no conflict of interest.

See the funding table on p. 15.

transported from the Earth would result in a large budget. For example, the United States Space Launch System is able to transport 70–130 tons of materials into Earth–Moon transfer orbit each time, but the launch cost could be ~$11.8 billion (1). If physical and chemical methods are used to carry out *in situ* regeneration, the required devices and consumables are difficult to regenerate and maintain *in situ* due to the lack of industry in the space environment. After billions of years of evolution, organisms have obtained efficient *in situ* regeneration and self-maintenance capabilities. Therefore, long-term crewed missions must adopt a bioregenerative life support system (BLSS), which is an artificial and closed ecosystem operated based on the principle of terrestrial ecosystem (2). BLSS is able to regenerate and circulate life support materials and dispose of waste *in situ* using biological methods, which is especially valuable for long-term space missions. We successfully completed the world's longest continuous crewed BLSS experiment ("Lunar Palace 365" experiment) in the facility of "Lunar Palace 1," a large-scale Earth-based closed experimental facility of BLSS (3).

In BLSS, the importance of waste treatment is almost equal to that of life-support materials' production, because there is no real "waste" in such closed-cycling and small-scale systems, which makes the waste disposal equivalent to resource production. In waste treatment, the treatment of organic solid waste involves a large amount of elements and their return to the BLSS material cycle (4). Due to its complexity and recalcitrance for degradation, organic solid waste is a key restricting and rate-limiting step of the material cycle (5). Discarding it directly to space would lead to pollution in the space environment (not conducive to planetary protection) and a considerable loss of potential life-support materials. Organic solid waste mainly comes from inedible plant waste (of ~90% dry weight) and the astronauts' feces (of ~10% dry weight), with lignocellulose as the major chemically recalcitrant component (6). The solid waste's biological degradation method in BLSS is mainly based on microbial aerobic fermentation, because anaerobic fermentation may produce toxic gases such as $NH_3$ and $H_2S$ which also lead to the loss of nutrient elements (H, N, S, etc.) (7, 8).

BLSS deployed in the space environment will face challenges from harsh environmental factors such as microgravity/low gravity, radiation, weak magnetic field, high vacuum, and large temperature differences (2). Space BLSS should not simply copy the conclusions and technologies obtained from Earth-surface-condition experiments such as "Lunar Palace 365," but needs to be corrected according to the experimental conclusions in the space environment or simulated space environment (9). Among these environmental factors, microgravity/low gravity is ubiquitous, difficult to shield, and with significant biological effects (9). As a force that acts directly on organisms, gravity is an important biological tropism factor and also largely affects other environmental factors such as water distribution, air convection, elasticity (the materials' compactness), and nutrient/toxin mass transfer (10). Therefore, microgravity/low gravity is one of the most important environmental factors to be studied for the realization of space BLSS. In space, there are both microgravity ($\leq 10^{-4}$ g, the critical acceleration of the biological tropism's gravity response [11]) and low-gravity environments (about 1/6 g on the Moon, and about 3/8 g on Mars). But microgravity ($\mu$-g) is the extreme case of gravity conditions, whose conclusions could be more valuable for scientific research, and the corresponding correction methods would be stronger as well. Therefore, we chose simulated microgravity (m$\mu$-g) as the environmental condition object for this study. Clinostats are one of the most feasible systems for simulating the biological effects of microgravity on the Earth, placing organisms in frames connected by one or two rotating axes that rotate at random or specific angular speeds (usually 1.5°–15° per second and ≤60° per second). The direction of gravity on the organism is ever-changing to achieve the microgravity vector (11).

Existing studies are abundant on both lignocellulosic organic solid waste degradation by microbial communities and the effect of microgravity on microbial pure cultures' physiology, but there are few studies focusing on the combination of them (12). In this study, wheat straw (as a lignocellulosic organic solid waste) was treated (degraded) by

microbial communities (the inoculum was the solid waste from the "Lunar Palace 365" experiment) for 25 days under mμ-g by clinostats or conventional 1 g conditions. We comprehensively analyzed the material's composition, microbial community composition and internal interactions, and metabolomics information. Based on the results obtained, we proposed a hypothesis: as a complex macromolecule, the degradation of lignocellulose is a multistep and highly coordinative task for the microbial community, and mμ-g confined the buoyant convection of both microbial cells and their metabolites, which broke the linkages among the degradation steps, thus fragmenting the functional microbial community; the caused local resource shortage also intensified microbial internal conflicts. Compared to previous studies' focus on microbial single species and their physiology under μ-g/mμ-g, this study tries to illustrate mμ-g's confinement and fragmentation on the microbial community and its multistep and highly coordinative degradation tasks, and thus serve as a "space" calibration to the method parameters obtained from Earth-based BLSS experiments, which is mandatory for future long-term crewed space missions.

## RESULTS

### Fermentation material's physical and chemical properties

The fermentation material's compositional degradation rates and possible related physical/chemical properties were measured. The mμ-g group's total material weight decreased significantly slower ($P < 0.05$) than the 1 g group since day 5 of the fermentation (Fig. 1a), showing a lower straw degradation rate. After day 20, their degradation rates became closer (Fig. 1a). Finer observations showed that, among the straw's three major macromolecular components, cellulose contributed the most to the mμ-g's slower degradation ($P < 0.05$), and hemicellulose degradation was not affected by mμ-g (Fig. 1b). Despite its low absolute content (about 1/20 of cellulose), lignin's degradation was even more suppressed by mμ-g ($P < 0.01$, Fig. 1c). As expected, the degradation progress visualized by the straw cell wall's autofluorescence on the fermentation's final day (day

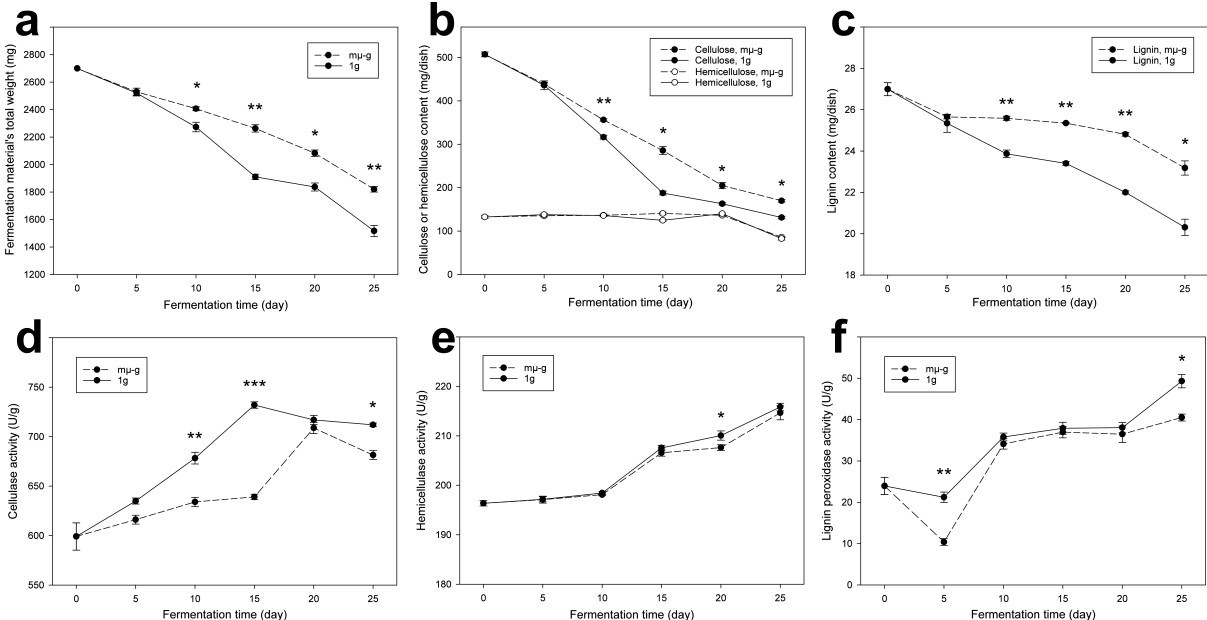

**FIG 1** The straw fermentation material's weight loss under mμ-g and 1 g, including (a) the material's total weight, and its three major lignocellulosic components of (b) cellulose and hemicellulose, and (c) lignin; and the material's lignocellulose degrading enzyme activities under mμ-g and 1 g, including (d) cellulase, (e) hemicellulase, and (f) lignin peroxidase. Error bars represent standard error ($n$ = 3 biological replicates). Asterisks indicate significant differences between mμ-g and 1 g, as determined by analysis of variance (ANOVA) (significance marks: *, $0.01 \leq P < 0.05$; **, $0.001 \leq P < 0.01$; ***, $P < 0.001$).

25) showed more blurred cell boundaries under 1 g (Fig. S2), indicating its lignocellulosic cell wall's faster degradation than that under mμ-g.

Consistent with the three major macromolecular components' degradation rates (Fig. 1), their degrading enzymes showed significantly lower activities ($P < 0.05$) under mμ-g for cellulase and lignin peroxidase, but not for hemicellulase (Fig. 1d through f). Physical properties of the fermentation material were not obviously affected by mμ-g, including conductivity, moisture, and pH (Fig. S3). The conductivity was generally higher under 1 g (Fig. S3a), likely due to the larger amount of ions released by stronger degradation. The pH increased substantially from neutral to alkaline (almost ~11, $P < 0.001$, Fig. S3c), suggesting ammonia production from the protein utilization as a carbon source and/or consumption of organic acids by the fast-growing microbial communities.

## Microbial communities

To explore the mechanisms of mμ-g's influence on fermentation via microbial communities, 33 fermentation material samples (= 2 treatment groups × 5 time points × 3 biological replicates + 3 time-zero samples) underwent amplicon sequencing and generated an average of 45,377 non-chimeric amplicon sequences for bacterial 16S rRNA gene and 64,838 for fungal ITS region, which were clustered into 2,101 bacterial amplicon sequence variants (ASVs) and 6,156 fungal ASVs. By the annotation, bacteria were classified into 452 species, and fungi were classified into 1,008 species.

In our study, fungi (~400 ASVs per sample, Fig. S4a) showed a much higher richness than bacteria (<200 ASVs per sample, Fig. S4b) in both 1 g and mμ-g conditions, given the fungi's well-characterized central role in lignocellulose degradation (13). Notably, the overall Shannon diversity index of fungi significantly reduced under mμ-g ($P < 0.05$, paired Student's $t$-test, Fig. S4c), which may partially explain the suppressed degradation of straw under mμ-g conditions (Fig. 1). In terms of microbial community composition, bacterial taxonomic annotation showed dominance by three phyla: *Actinobacteria*, *Firmicutes,* and *Proteobacteria*, with *Proteobacteria* gradually increasing in relative abundance over time (from ~0% to ~80%, Fig. S5a). Meanwhile, fungal communities were consistently dominated by the phylum *Ascomycota* (Fig. S5b).

To deeply explore functional interactive patterns within lignocellulose-degrading communities, two phylogenetic molecular ecological networks (pMENs) with fungi, bacteria, and other related factors were constructed, respectively, under 1 g and mμ-g (Fig. 2). As expected, both networks showed topological properties of small world, scale-free, and modularity (14, 15), and were significantly different from randomly generated networks (Table S1), validating their representativeness for complex systems' networks. Fungal nodes were overwhelmingly more than bacterial in both networks, verifying their much deeper involvement in lignocellulose degradation; nevertheless, a "fungal module" and a "bacterial module" are recognized in both networks (Fig. 2). Notably, the mμ-g treatment decreased the network size by 46% and the connectivity by 60%, and increased the modularity by 67% (Table S1), seemingly adverse to the high functional connectivity and involvement required for the degradation of recalcitrant macromolecules. Consistently, the data indicate that mμ-g resulted in a much more fragmented and sparser network (Fig. 2b), implying its fragmenting effect on microbial communities.

## Metabolomics analysis

To deeply illustrate the underlying mechanisms, the material's metabolites and potential pathways were measured and analyzed. The principal coordinate analysis (PCoA) of the non-targeted metabolomics result showed that 1 g and mμ-g are distinguished on days 5, 15, and 20 of the fermentation (Fig. S6), and we found the metabolites of days 5, 10, and 25 lacking annotation information in mainstream databases such as Kyoto Encyclopedia of Genes and Genomes (KEGG) or hardly hitting microbial-related pathways (Table S2), so we performed finer observations on days 15 and 20. Firstly, it is notable that mμ-g significantly ($P < 0.05$) enriched many antimicrobial metabolites (Fig. 3a), including

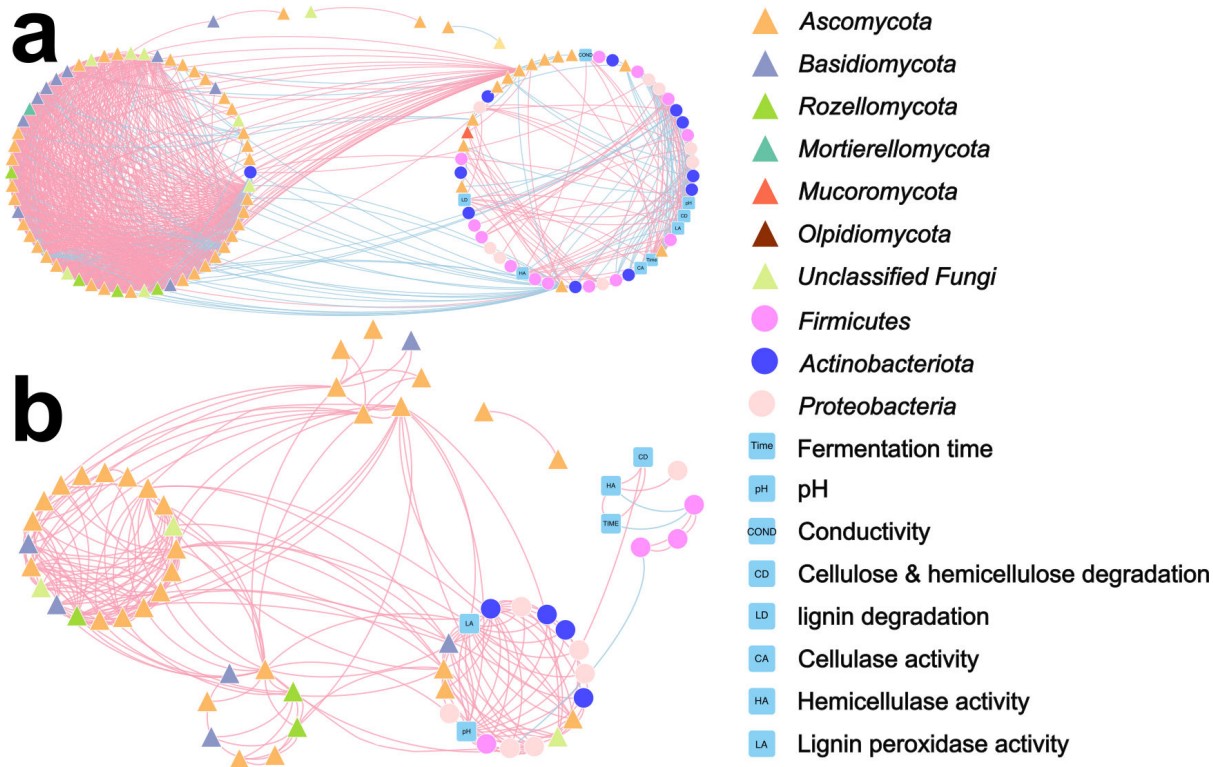

**FIG 2** Molecular ecological networks among the fermentation material's microbial communities and its physical/chemical/enzymatic properties, including the networks of (a) 1 g and (b) mμ-g group. Triangular nodes represent fungi, and circular nodes represent bacteria (both indicated in the level of phylum). Red links indicate positive correlations between two nodes, while blue links indicate negative ones.

reported antifungal chemicals cyclohexylamine (increased by 188 times), L-norvaline, and pelletierine, and broad-spectrum antimicrobial chemicals p-tert-amylphenol, (R)-(+)-alpha-terpineol, and trypanothione disulfide, suggesting increased conflicts under mμ-g, especially against fungi. Consistently, several reported fungal (antibacterial) metabolites were reduced by mμ-g, including paxilline (by *Penicillium paxilli*) and fumigaclavine B (by *Aspergillus*), which echoes the decreased fungal diversity (Fig. S4c). In addition to fungi, considering another efficient lignocellulose degrading phylum—the *Actinobacteria* of bacteria, a typical metabolite gentamicin X2 of the genera *Micromonospora*, and two reported metabolites of *Streptomyces*, pentalenolactone D and acetyldemethylphosphinothricin tripeptide, were also significantly reduced by mμ-g (Fig. 3a), suggestive of a confined activity of *Actinobacteria*.

Some plausible downstream products of lignin degradation were reduced by mμ-g, such as benzene, 7-O-galloyl-D-sedoheptulose and (2,3-dihydroxybenzoyl)adenylate (Fig. 3a), which is in concert with the suppressed lignin degradation (Fig. 1c). Some straw degradation products were also reduced by mμ-g, such as octadecadienoic acid amide (from linoleic acid), suggesting the slowed-down degradation efficiency. N-(3-hydroxy-7-cis-tetradecenoyl)homoserine lactone, a quorum sensing signal molecule of *Rhizobium leguminosarum*, was reduced under mμ-g, implying a confined bacterial quorum sensing ability.

We mapped differential metabolic modules from the data of day 15 and day 20 using iPath3.0. The mμ-g significantly reduced the degradation of polycyclic aromatic hydrocarbons (PAHs, $P < 0.05$, Fig. 3b), suggesting that microbes were busy coping with toxic volatile organic compound (VOC) stress generated by other microbes (given that PAHs are typically not from lignocellulosic degradation). However, the degradation of many mono-aromatic hydrocarbons, such as benzene, benzoic acid, vanillin, cis-benzeneglycol, and other real downstream products of lignin degradation was reduced at mμ-g ($P <$

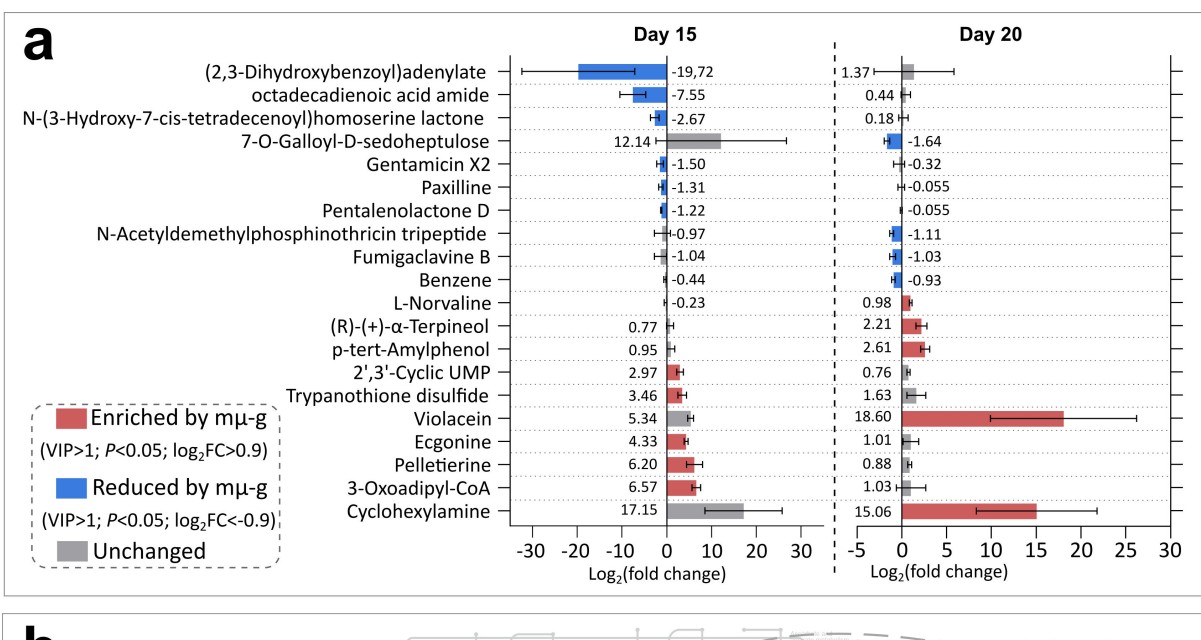

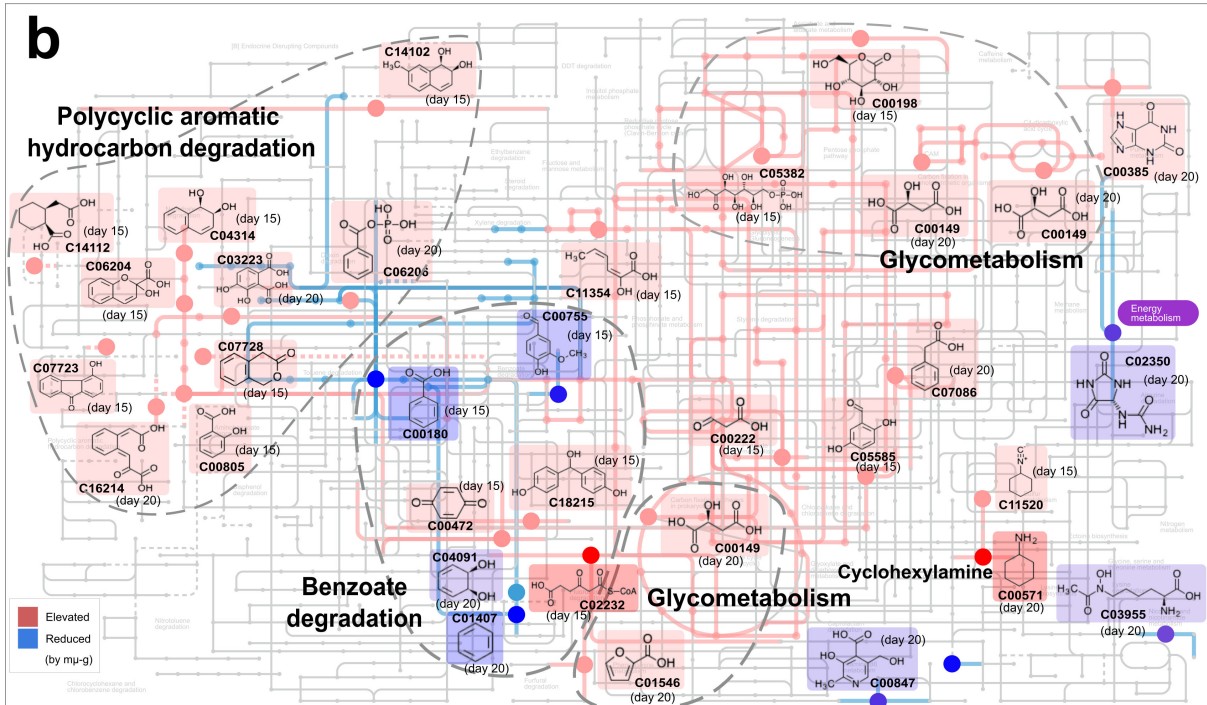

**FIG 3** Metabolomics of the fermentation material on day 15 and day 20 (results pooled together with the time points labeled) showing significant enrichments and reductions under mμ-g (VIP > 1, $P < 0.05$, |log$_2$FC| > 0.9), which include (a) metabolites, error bars represent standard error ($n = 3$ biological replicates); numbers indicate the log2 fold change (log$_2$FC) values for each metabolite and (b) metabolic modules located on the map of xenobiotics biodegradation and metabolism. Only representative and abundant metabolites are shown in (a), or highlighted in (b).

0.05, Fig. 3b), echoing the weight loss difficulties of lignin (Fig. 1c). The pentose phosphate pathway of glucose metabolism was elevated by mμ-g ($P < 0.05$, Fig. 3b), suggesting the microbial community's tendency to anabolism rather than energy metabolism, as well as a tendency for stress resistance.

Given this and the importance of glycometabolism for cellulose degradation and microbial physiology, we examined the metabolites in the three major pathways of glycometabolism. We found that mμ-g increased the contents of many intermediate metabolites, including glyceraldehyde 3-pi and 3-pi-glycerate in the glycolytic pathway

(day 20), L-malate and isocitrate in tricarboxylic acid cycle (day 20), and sedoheptulose-7-pi and erythrose 4-pi in pentose phosphate pathway (days 15 and 20) ($P < 0.05$, Fig. S7), suggesting an accumulation of multiple intermediate metabolites and thus a confined glycometabolism under mμ-g.

## The visualizing investigation of mμ-g's physical effect using *Aspergillus nidulans*

We tried to partly visualize and explore the mechanism of the mμ-g's confining and fragmenting physical effect on microbes by the pure culture of *Aspergillus nidulans* (strain purchased) grown on cellulose—Congo Red solid plate media, as the species was a dominant lignocellulose degrading fungus in the current study (averagely 22.0% of the overall fungal abundance). As the straw experiment was also done on the wet but solid straw material, we suppose such a solid-plate-media pure-culture experiment could be comparable and simplifying. The result showed that mμ-g obviously decreased the colonies' total size (Fig. 4a, and the increased colony number likely resulted from the unexpected "secondary inoculation" by the clinorotation), suggesting a confined colony-size expansion ability under mμ-g. Clear zones on the Congo-Red-added solid plate media, showing the degradation of cellulose, were also more obvious and wider under 1 g (Fig. 4a).

Microscopic observation on the colonies showed that mμ-g obviously increased the density of *Aspergillus nodulans*' ascomata (Fig. S8), significantly increasing the number of ascomata per unit area by 46.9% ($P < 0.05$, Fig. 4b), which likely verified the mμ-g's expansion-confining mechanism on *A. nodulans*' hyphae (the vegetative organ of fungi) and colonies microscopically. We hereby illustrate this convection-based model of the mμ-g's confinement and fragmentation on lignocellulose degrading microbial community by a schematic diagram (Fig. 5).

## DISCUSSION

Effects of microgravity (μ-g) and mμ-g on microbial growth and secondary metabolism have been studied for more than 50 years (12). These studies have been focusing on single microbial species/strains living on prepared pure media under μ-g/mμ-g, leaving microbes presented as communities or living on natural and chemically recalcitrant substrates understudied, which are actually closer to practical application scenarios in crewed spacecraft and extraterrestrial bases. The current study hence focuses on

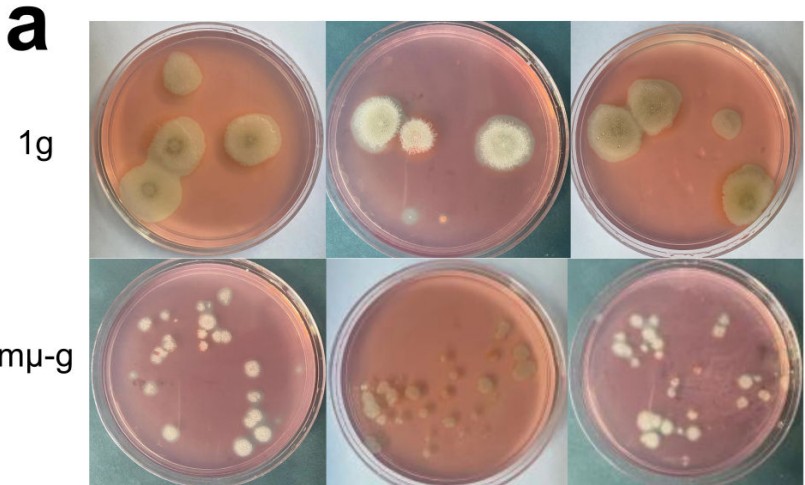 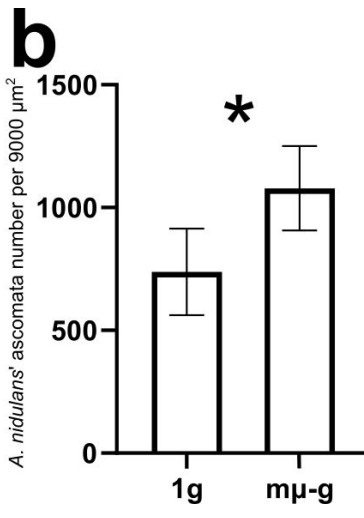

**FIG 4** The pure culture of *Aspergillus nidulans* grown on cellulose—Congo Red solid plate media, which visualized and explored the mechanism of the mμ-g's confining and fragmenting physical effect on microbes, including (a) the plates' macroscopic photographs, and (b) the number of ascomata per unit area from microscopic observation. Asterisks indicate significant differences determined by ANOVA (significance marks: *, $0.01 \leq P < 0.05$, $n = 15$).

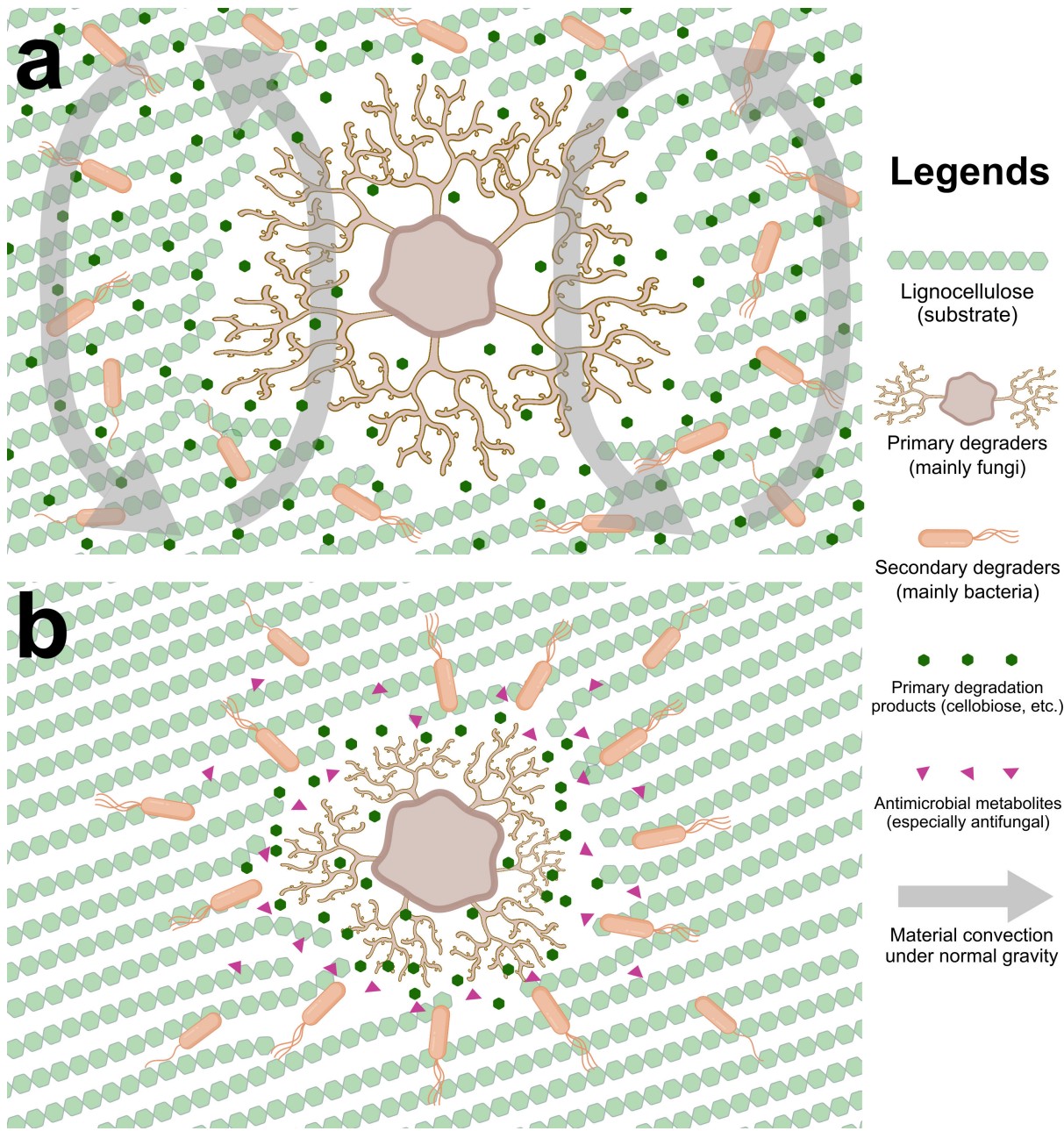

**FIG 5** The schematic diagram of the material-convection-based lignocellulose degradation model under different gravitational conditions proposed by this paper shows (a) the degradation under 1 g which has a normal material convection of the primary degradation products, and (b) a confined and fragmented convection under mμ-g and the derived local resource shortage, which causes the zonal concentration of microbes and thus intensified conflicts manifested in the release of antimicrobial metabolites, especially against fungi, the principal primary degrader.

such scenarios, trying to provide an important supplementary perspective to space microbiology theoretically, and also a practical basis for lignocellulose-based solid waste treatments in crewed long-term space missions (8). This study simulated microgravity by clinostats, whose results have been shown as similar to real microgravity and also thoroughly compared to other simulators on the ground (12).

Gravity is the basic driving force of the buoyant form of thermal and material convections, while μ-g/mμ-g conditions have no or very low gravity, which would thus largely confine buoyant convections (16). Such a material-convection-mediated indirect impact could be the dominant way for microbes to perceive μ-g/mμ-g, as they are

simple lifeforms without gravireceptors, unlike higher plants and animals (12). In the current study's scenario, the degradation of recalcitrant macromolecules like lignocellulose is a multistep and highly coordinative task for its functional microbial community, in which each step generates different products as the next step's substrate (17), and may thus highly rely on material convection of microbial cells, their substrates/products, and corresponding extracellular enzymes. Therefore, μ-g/mμ-g may fragment the microbial cooperative degradation chains/networks based on convection. As expected, mμ-g significantly slowed down weight loss rates of the straw fermentation's total material, cellulose, and lignin (Fig. 1), also further validated by their degrading enzymes' lower activities ($P < 0.05$) under mμ-g for cellulase and lignin peroxidase (Fig. 1d and f). Molecular ecological network analysis revealing microbial interactions showed that mμ-g decreased the network size by 46% and the connectivity by 60%, and increased the modularity by 67% (Fig. 2; Table S1), which is in concert with the hypothesis that mμ-g may convectively fragment the microbial interactive chains/networks needed for lignocellulose degradation, excluding many microbes that should have been involved.

The pure-culture visualization experiment further echoes this hypothesis. The solid-plate-media pure culture of *Aspergillus nidulans* suggested a largely confined colony-size expansion ability under mμ-g (Fig. 4a), and the microscopic observation showed significantly higher ascomata densities and thus shorter hyphal extension ranges (Fig. 4b; Fig. S8). Hypha is the vegetative organ of fungi (18), which should be directly involved in lignocellulose degradation and assimilation. This result further corroborated the mμ-g's physical confinement on fungal diffusion and expansion. In the case of the lignocellulose's multistep and highly coordinative degradation task (17), such confinement on microbial cells, their metabolic products/substrates, and extracellular enzymes may result in local resource shortages, as well as the zonal concentration of next-step degraders and thus intensified conflicts (Fig. 5).

Consistently, mμ-g significantly ($P < 0.05$) enriched many antimicrobial metabolites (Fig. 3a), including the reported antifungal chemicals cyclohexylamine (increased by 188 times) (19), L-norvaline (20) and pelletierine (21), and broad-spectrum antimicrobial chemicals p-tert-amylphenol (22), (R)-(+)-alpha-terpineol (23), and trypanothione disulfide (24), suggesting increased microbial conflicts under mμ-g manifested in chemical release, especially against fungi (the dominant lignocellulose degrader). Metabolic module analysis also showed that mμ-g significantly enriched the degradation of polycyclic and condensed aromatics ($P < 0.05$, Fig. 3b), which are typically not from the three monocyclic monomers of lignin degradation (coniferyl, sinapyl, and p-coumaryl alcohols) and likely toxic to microbes, while the real, monocyclic products of lignin degradation (such as benzoic acid) were reduced under mμ-g (Fig. 3b). These results together echo the pattern of microbial community internal conflicts, resisting the chemicals synthesized and released by other microbes.

Similar convection-based models of microbial growth and secondary metabolism under mμ-g have previously been proposed based on bacterial single-species liquid-culture experiments, also emphasizing the mμ-g's impact on material convections rather than directly on the cells (10), and hypothesizing the zonal accumulation of metabolic products surrounding the cells (25). Such pure liquid culture experiments usually observe higher bacterial biomasses under μ-g/mμ-g (25, 26), but most of them adopted chemically labile substrates, instead of recalcitrant macromolecular ones whose utilization could be coordinative tasks. In comparison, as the first-step degraders of lignocellulose (17), fungi were largely suppressed under mμ-g in the current study (Fig. S4c; Fig. 3a and 4), which may serve as the bottleneck for overall biomass accumulation. This study focuses rather on a much more complex and practical scenario of multistep degradation, which is community-based, oligotrophic, competitive, mixed-phase (physically), and fungi-dominant. Compared to the pure culture's simple pattern, such a harsh scenario's degradation efficiency typically requires a more "globalized" interaction within the community. Nevertheless, under mμ-g, the slowed-down overall degradation rate (Fig. 1), the fragmented microbial community networks (Fig. 2), the intensified

antimicrobial metabolites release (Fig. 3), and the visualized confined expansion of the principal degrader (Fig. 4) all suggest a confining, fragmenting, and isolating role of mμ-g on lignocellulose degraders, intensifying their local resource shortages and internal conflicts (Fig. 5). Therefore, this study likely extended the hypothesized convection-based model's specific effect to lignocellulose microbial degradation. Proper mass-transfer-promoting methods should be adopted for the "globalization" of microbial interactions required by the lignocellulose degradation efficiency in space, such as stirring, heat gradient, moisture gradient, etc.

Beyond this hypothesized model, some other findings are also worth discussing. It is notable that mμ-g largely suppressed the rate of lignin weight loss (Fig. 1c), and the activity of lignin peroxidase as well (Fig. 1f). Despite its relatively low content in lignocellulose (Fig. 1), lignin usually serves as the physical barrier resisting enzymatic degradation in straws of crops such as wheat (27). Moreover, lignin may chemically inhibit the activities of cellulase, xylanase, and β-glucosidase, which highly determine lignocellulose's enzymatic degradation (28). Therefore, the suppressed lignin degradation we observed may also partly explain the suppressed cellulose and total weight loss (Fig. 1). We also found a likely suppressed activity of *Streptomyces* for its reduced metabolites pentalenolactone D (29) and acetyldemethylphosphinothricin tripeptide (30) (Fig. 3a), which is consistent with previous studies finding lower dry cell weights of two *Streptomyces* species, respectively, under rotating wall bioreactors (31) and high-aspect rotating vessels (32) (two different methods of mμ-g).

The application scenario of μ-g organic solid waste degradation should be in BLSS deployed in spacecraft and extraterrestrial bases, which require closed material cycling (3). Following this principle, the treated solid waste would subsequently go to the higher plant (crop)'s cultivation substrate or fertilizer, which puts forward higher demands on the treated waste's compatibility with plants (33). An issue of concern is the stubbornly increasing pH (from neutral to almost 11) of the fermented straw (Fig. S3c), likely too alkaline for plant cultivation. In such aerobic waste fermentations, microbial carbon consumption needs are large and quick, which makes the nitrogen source utilized as a carbon source, causing very active ammoniation and mineralization of organic nitrogen (34). This process could be the major driving force of the observed pH increase (Fig. S3c). Despite the weight-loss-rate advantage of simple aerobic fermentation, more natural and plant-friendly methods should be considered, such as the earthworm's treatment which typically ameliorates the pH (35). In terms of plant probiotic/pathogenic microbial community composition, *Proteobacteria* gradually took over bacterial abundance over time (from ~0% to ~80%), replacing *Actinobacteria* and *Firmicutes* (Fig. S5a). This could be plant-friendly because *Proteobacteria* is abundant in plant probiotics, such as the genus of *Rhodobacter* (36, 37), *Klebsiella* (38), *Delftia* (39), and *Caulobacter* (40); while *Actinobacteria* and *Firmicutes* are abundant in cellulolytic bacteria (41), which could threaten the plant's lignocellulosic cell wall and thus the root turgor pressure and apoplast pathway (42). Therefore, such a succession trend of bacteria during the fermentation could be friendly to the subsequently cultivated plants.

Studies on microbes under microgravity have been abundant, but showed plausible conflicting and diverse results, likely due to the "indirect" manner by which μ-g/mμ-g affect microbes, via intermediate factors such as microbial species/strain types, cell motility, culture methods (suspension or agar cultures), nutrition availability, and culture time (12). Therefore, the current study's complicated and unique scenario leads to the difficulty interconnecting with previous studies. An interesting "pure-culture network" experiment (43) may inspire the follow-up mechanistic study of the current one: a lignocellulose-based pure culture of the representative species under mμ-g, including each step's degraders and antimicrobial metabolite producers, could verify our conclusions and explore deeper.

In conclusion, the mμ-g slowed down the overall degradation rate of lignocellulose-based organic solid waste (Fig. 1), fragmented the microbial community networks (Fig. 2), and intensified the internal antimicrobial metabolites release (Fig. 3), which could

be partly explained by the confined expansion of *Aspergillus nidulans*, the principal degrader (Fig. 4). We hence proposed a model (Fig. 5): the low material convection under mμ-g may confine the diffusion of microbial cells, their metabolic products/substrates, and extracellular enzymes, thus fragmenting the microbial interactions needed for such a multistep and highly coordinative task of lignocellulose degradation; the caused local resource shortage may result in a zonal concentration of microbes and thus intensified conflicts, especially against fungi, the principal degrader. This convection-based model explains the observed phenomena and suggests proper mass-transfer-promoting methods for more "globalized" microbial interactions in such a community-based, oligotrophic, mixed-phase (physically), and fungi-dominant scenario. The higher lignocellulose degradation efficiency thus achieved would certainly improve the BLSS required for long-term space exploration missions.

## MATERIALS AND METHODS

### Straw degradation under simulated microgravity (mμ-g) and normal gravity (1 g)

The fermentation substrate in this study was air-dried wheat straw whose moisture content was <5%. The straw was ground by the FW100 grinder (Taisite, Tianjin, China) with 10 g of material added each time and ground for 30 s under the first gear. The ground material passing the 20-mesh sieve (particle size: 850 μm, Haiji Co., Ltd., Hengshui, Hebei, China) was collected for the experiment. The inoculum was the organic solid waste produced by our "Lunar Palace 365" experiment (3). Its major components were aerobic ferments of inedible plant parts (such as straw, old leaves, and plant roots), human (crew) feces, and their accompanying microbes (5). To make the inoculum more representative, we mixed solid waste samples collected on 4 July 2017 (~the initial phase of the "Lunar Palace 365" experiment) and on 15 May 2018 (~the final phase). Due to the high water content and large particles in the solid waste, to avoid insufficient contact with the crushed wheat straw, it was air dried for 48 h at 30°C until the water content reduced to less than 5%, and then ground to powder. Because air drying would be a typical treatment for straw and solid waste (3), the air-drying method in this study conforms to the application scenario's effect on the material and its microbial community.

According to the mixing ratio with the highest degradation weight loss rate established by us earlier (8), water, straw, and solid waste inoculum were mixed and fully stirred in a clear ziplock bag with a ratio of 65:35:7, and each 2.7 g was weighed and placed in a small petri dish (35 mm × 10 mm) as a biological replicate (*n* = 3). Petri dishes are without sealing film to ensure aerobic conditions and thus material exchange with outside, and deployed on the clinostats (Yulei Technology Co., Ltd., Tianjin, China). The clinostats were put into a climate chamber (HDL apparatus, Beijing, China) with a temperature of 45°C and humidity of 40% (8) to start the fermentation. The fermentation was carried out for a total of 25 days, and the fermented materials were sampled every 5 days (six-time points in total, including day 0). All samples are stored in a −80°C refrigerator for subsequent testing.

Two-dimensional (2-D) clinostats are widely applied to study microgravity effects on biological samples, which rotate perpendicularly to the direction of the gravity vector. Studies have shown that the results obtained from various model systems using 2-D clinorotation were similar to those found under real microgravity conditions (12, 44). To create the mμ-g condition, the clinostat was set to rotate at the speed of 1 rpm around the horizontal axis in a counterclockwise direction; the 1 g condition was set as identical but around the vertical axis (Fig. S1).

## Physical and chemical properties, lignocellulose contents, and enzyme activities

The moisture content of fermentation material was measured by the MB23 moisture content meter (OHAUS, Parsippany, New Jersey, USA). The fermentation material extract's pH was detected with the FE28 pH meter (Mettler Toledo, Zurich, Switzerland), with 2 g of fermentation material added to 15 mL of water (45). The conductivity of the extract was measured using the FE38 conductivity meter (Mettler Toledo). The material's contents of cellulose, hemicellulose, and lignin were measured by corresponding measurement kits (BOXBIO, Beijing, China); cellulase, hemicellulase, and lignin peroxidase activities' measurement kits (BOXBIO) were used to measure enzyme activities, both with 0.1 g of air-dried fermentation material for each measurement.

We observed wheat straw cell walls on the final day (day 25) of fermentation as a visualization of degradation progress, utilizing the cell wall's autofluorescence. To ensure the representability, 0.2 g of fermentation material from each of the three replicates was thoroughly mixed in 15 mL of sterile water, and 1 mL of suspension was absorbed and placed on the microscopic slide. The fluorescence microscopy BX43F (Olympus, Tokyo, Japan) and its 100-W mercury lamp power box U-RFL-T (Olympus, Tokyo, Japan) were used for fluorescence imaging with excitation light in the wavelength range of 455–495 nm. The image is captured using the ImageView software.

## Sequencing analysis on the fermentation's microbial community

DNA extraction, PCR amplification, and sequencing of rhizospheric microbial communities were performed by Biomarker Tech. Corp., Beijing, China. Briefly, total genomic DNA was extracted from 0.5 g fermentation samples using the TGuide S96 Magnetic Soil/Stool DNA Kit (Tiangen Biotech [Beijing] Co., Ltd., China) according to the manufacturer's instructions. DNA concentration was measured with the Qubit dsDNA HS Assay Kit and Qubit 4.0 Fluorometer (Invitrogen, Thermo Fisher Scientific, Eugene, Oregon, USA). The V3–V4 region of 16S rRNA genes and the fungal ITS1 region were amplified using primer pairs (forward of 5′ -ACTCCTACGGGAGGCAGCA-3′ and reverse of 5′ -GGACTACHVGGGTWTCTAAT-3′ for 16S rRNA genes; forward of 5′ -CTTGGTCATTTAGAG GAAGTAA-3′ and reverse of 5′ -GCTGCGTTCTTCATCGATGC-3′ for ITS1 region). Both the forward and reverse primers were tailed with sample-specific Illumina index sequences. The PCR was performed in a total reaction volume of 10 µL: DNA template 5–50 ng, *Vn F (10 µM) 0.3 µL, *Vn R (10 µM) 0.3 µL, KOD FX Neo Buffer 5 µL, dNTP (2 mM each) 2 µL, KOD FX Neo 0.2 µL, and ddH$_2$O up to 10 µL. Vn F and Vn R were selected according to the amplification area. The initial denaturation at 95°C for 5 min was followed by 25 cycles of denaturation at 95°C for 30 s, annealing at 50°C for 30 s and extension at 72°C for 40 s, and a final step at 72°C for 7 min. PCR amplicons were purified with Agencourt AMPure XP Beads (Beckman Coulter, Indianapolis, IN, USA) and quantified using the Qubit dsDNA HS Assay Kit and Qubit 4.0 Fluorometer (Invitrogen, Thermo Fisher Scientific). The quantified amplicons were then pooled together in equal amounts as the library. The constructed library was sequenced using Illumina NovaSeq 6000 (Illumina, Santiago, CA, USA).

Downstream sequencing analysis was performed on BMK Cloud (Biomarker Technologies Co., Ltd., Beijing, China). Raw sequences were first processed using Trimmomatic and FLASH, with a moving window of 50 bp and a quality threshold score of 30. Singletons were then removed. Next, high-resolution ASVs were identified from the reads using DADA2 (version 1.26) (46). Lastly, a representative sequence of each bacterial ASV was annotated through the SILVA ribosomal RNA gene database (version 138) and each fungal ASV was annotated through the UNITE database both with a confidence score of 0.7 (47).

## Non-targeted metabolomics

Non-targeted metabolite profiling of fermentation samples was conducted by Biomarker Technologies Corporation, Beijing, China. The fermentation samples (50 mg) were freeze-dried and ground into a fine powder. The powdered samples were then dissolved in 1,000 µL of extraction solvent (methanol:acetonitrile:water = 2:2:1, vol/vol/vol) containing an internal standard (2-chloro-L-phenylalanine, 20 mg/L). The mixture was vortexed for 30 s, homogenized using a grinder at 45 Hz for 10 min, and sonicated in an ice-water bath for 10 min. After incubation at −20°C for 1 h, the samples were centrifuged at 12,000 rpm for 15 min at 4°C. The supernatant (500 µL) was collected, dried using a vacuum concentrator, and reconstituted in 160 µL of reconstitution solvent (acetonitrile:water = 1:1, vol/vol). The final extract was centrifuged again, and 120 µL of the supernatant was transferred to injection vials for liquid chromatography-mass spectrometry analysis. Each sample was combined with 10 µL from other samples to create a quality control sample for machine analysis.

Metabolite analysis was performed using a Waters Acquity I-Class PLUS ultra-high-performance liquid chromatography (UPLC) system coupled with a Waters Xevo G2-XS QTOF high-resolution mass spectrometer. Chromatographic separation was achieved using a Waters Acquity UPLC HSS T3 column (1.8 µm, 2.1 × 100 mm). The mobile phase consisted of 0.1% formic acid in water (mobile phase A) and 0.1% formic acid in acetonitrile (mobile phase B) for both positive and negative ion modes. The gradient program was as follows: 0–0.25 min, 98% A; 0.25–10 min, linear gradient to 2% A; 10–13 min, 2% A; 13.1–15 min, return to 98% A. The flow rate was maintained at 400 µL/min. Mass spectrometry data were acquired in MSe mode, with low collision energy set at 2 V and high collision energy ranging from 10 to 40 V. The scanning frequency was 0.2 s per spectrum. The ESI ion source parameters were as follows: capillary voltage at 2,000 V (positive ion mode) or −1,500 V (negative ion mode), cone voltage at 30 V, ion source temperature at 150°C, desolvation gas temperature at 500°C, backflush gas flow rate at 50 L/h, and desolvation gas flow rate at 800 L/h. The mass-to-charge ratio (m/z) range was set to 50–1,200.

The raw data collected using MassLynx V4.2 software were processed using Progenesis QI software for peak extraction, peak alignment, and other data processing operations. Metabolite identification was performed using the METabolite LINkage Database, KEGG, Human Metabolome Database, and Lipid Metabolites and Pathways Strategy databases. Theoretical fragment identification was conducted with a mass deviation within 100 ppm for precursor ions and 50 ppm for fragment ions. To preliminarily understand the overall metabolic differences among each group of samples, orthogonal projections to latent structures-discriminant analysis were conducted on metabolome data using the R software package "ropls" with a 200-time permutation to verify the model's reliability. A $t$-test was used to calculate significant differences between treatment groups for each metabolite. The VIP value of the model was calculated using multiple cross-validation. Differential metabolites were screened based on the criteria of $|\log_2(\text{fold change})| > 0.9$, $P < 0.05$, and VIP $> 1$. The differential metabolites of KEGG pathway enrichment significance were calculated using a hypergeometric distribution test (R package "clusterProfiler").

## Molecular ecological network analysis, statistical analysis, and graphing

With 16S rRNA gene (bacteria) ASVs, ITS ASVs, and other related factors pooled together as the input, pMENs were constructed based on a random matrix theory (RMT)-based network (15). The threshold of similarity coefficients (r values of the Spearman's rho correlation) for network construction was automatically determined when the nearest-neighbor spacing distribution of eigenvalues transitioned from Gaussian orthogonal ensemble to Poisson distributions (15). Environmental factors (including wheat and solid waste parameters) were also incorporated, as RMT-based networks allow multiple input data types (15). Random networks corresponding to all pMENs were constructed using the Maslov-Sneppen procedure with the same network size and average link number to

verify the system-specificity, sensitivity, and robustness of the empirical networks (48). Network graphs were visualized with Cytoscape 3.8 software.

Matlab (2020) and SigmaPlot 14.0 software were used to plot the curve plots. Analysis of variance (ANOVA) by the package "vegan" (version 2.3-2) in R software was used to examine significant differences, and the non-parametric paired Wilcoxon test was used for experimental data not conforming to normal distribution. The microbial Shannon index (α-diversity) was calculated and displayed by QIIME2 and R software (version 4.3.1). The β-diversity of microbial communities was calculated based on Bray-Curtis and UniFrac distances and visualized by PCoA. Metabolomics pathways, modules, and nodes were graphed using iPath3.0 (49). The schematic diagram was created with BioRender.com. If not otherwise stated, the mean ± standard error of $n = 3$ samples for the three fermentation technical replicates are shown.

## The solid-plate-media verification experiment using *Aspergillus nidulans*

As we found that *Aspergillus* was potentially responsible for straw degradation, the purchased strain of *Aspergillus nidulans* (China Agricultural Microbial Culture Preservation and Management Center, Beijing, China), a reported dominant cellulose-degrading fungus in *Aspergillus*, was selected as the experimental object of this part. Fifty milligrams of freeze-dried powder of the strain was completely dissolved in 5 mL of PBS buffer. The dissolved fungal solution was then diluted into a series of concentrations ranging from $10^{-1}$ to $10^{-7}$, and spread onto sterile carboxymethyl cellulose solid medium plates with 100 μL each. The plates were sealed and placed on 1 g and mμ-g clinostats and incubated under a temperature of 45°C and a humidity of 40% for 3 days. Based on the principle of Congo Red's red labeling on cellulose, 20 mL of Congo Red solution (mass fraction: 0.04%) was poured into each plate and stood for 15 min, and 15 mL of 1 mol/L NaCl solution was added to rinse for 15 min, after which the clear zones around the colonies showing the cellulose degradation situation could be observed. Plates with a dilution factor of $10^{-4}$, where the colony edges were clear and the number of colonies was moderate, were selected for photography. From the microscopic photos taken on the colonies (close to the colonies' margin, 2,592 × 1,944 pixels), the number of *A. nodulans*' ascomata was counted using ImageJ software. Specifically, we converted the image format to 8-bit; set the image-adjust-threshold (50–255), and binarized the image; segmented the overlapped ascomata by the function of Process-Binary-Watershed to make the counting more accurate; carried out automatic counting by the function of Analyze-Analyze Particles with the counting threshold of 30 pixel$^2$.

## ACKNOWLEDGMENTS

This work was financially supported by a grant from the National Natural Science Foundation of China (32200104), the Fundamental Research Funds for the Central Universities, and the Research Funding of Hangzhou International Innovation Institute of Beihang University (Grant no. 2024KQ100).

B.L., writing—review and editing, formal analysis, investigation, methodology, visualization | T.F., writing—original draft, formal analysis, investigation, methodology, data curation | S.H., data curation, formal analysis | J.F., conceptualization, funding acquisition, project administration, supervision, visualization, writing—review and editing | H.L., conceptualization, investigation, methodology, project administration, supervision | All authors read and approved the final manuscript.

## AUTHOR AFFILIATIONS

[1]Institute of Environmental Biology and Life Support Technology, School of Biological Science and Medical Engineering, Beihang University, Beijing, Beijing, China
[2]International Joint Research Center of Aerospace Biotechnology & Medical Engineering, Beihang University, Beijing, Beijing, China

[3]Innovation Center for Medical Engineering & Engineering Medicine, Hangzhou International Innovation Institute, Beihang University, Hangzhou, Zhejiang, China

## AUTHOR ORCIDs

Hong Liu  http://orcid.org/0000-0003-4343-3490
Jiajie Feng  http://orcid.org/0000-0002-5797-8400

## FUNDING

| Funder | Grant(s) | Author(s) |
|---|---|---|
| National Natural Science Foundation of China | 32200104 | Jiajie Feng |
| Fundamental Research Funds for the Central Universities | | Jiajie Feng |
| Research Funding of Hangzhou International Innovation Institute of Beihang University | | Hong Liu |

## AUTHOR CONTRIBUTIONS

Boyang Liao, Formal analysis, Investigation, Methodology, Visualization, Writing – review and editing | Tianyi Feng, Data curation, Formal analysis, Investigation, Methodology, Writing – original draft | Sihan Hou, Data curation, Formal analysis | Hong Liu, Conceptualization, Investigation, Methodology, Project administration, Supervision | Jiajie Feng, Conceptualization, Funding acquisition, Project administration, Supervision, Visualization, Writing – review and editing

## DATA AVAILABILITY

Raw 16S rRNA gene and ITS amplicon sequences are available in the NCBI SRA database (http://www.ncbi.nlm.nih.gov/sra) under study no. PRJNA1197767 and no. PRJNA1198645, respectively. Metabolomics data is available in Table S2.

## ETHICS APPROVAL

This article does not contain any studies with human participants or animals performed by any of the authors.

## ADDITIONAL FILES

The following material is available online.

### Supplemental Material

**Supplemental material (Spectrum02466-24-s0001.docx).** Tables S1; Fig. S1 to S8.
**Table S2 (Spectrum02466-24-s0002.csv).** Metabolomics raw data.

### Open Peer Review

**PEER REVIEW HISTORY (review-history.pdf).** An accounting of the reviewer comments and feedback.

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
