## [Reviewer comments · Microbiology Spectrum]

Microbiology Spectrum

Simulated microgravity confines and fragments the straw-based lignocellulose degrading microbial community

Boyang Liao, Tianyi Feng, Sihan Hou, Hong Liu, and Jiajie Feng

Corresponding Author(s): Jiajie Feng, Beihang University

Review Timeline:

Submission Date:	October 11, 2024
Editorial Decision:	December 4, 2024
Revision Received:	January 11, 2025
Editorial Decision:	February 19, 2025
Revision Received:	February 24, 2025
Accepted:	March 14, 2025

Editor: Jing Han

Reviewer(s): The reviewers have opted to remain anonymous.

Transaction Report:

DOI: <https://doi.org/10.1128/spectrum.02466-24>

Re: Spectrum02466-24 (Simulated microgravity confines and fragments the straw-based lignocellulose degrading microbial community)

Dear Dr. Jiajie Feng:

Thank you for the privilege of reviewing your work. Below you will find my comments, instructions from the Spectrum editorial office, and the reviewer comments.

Revision Guidelines

Sincerely,
Jing Han
Editor
Microbiology Spectrum

Reviewer #1 (Public repository details (Required)):

Sequencing and Metabolomics data both need to be included in a public repository

Reviewer #1 (Comments for the Author):

General Comments

- 1) One of the main conclusions from the authors seems to be that microgravity disrupts transport of diverse microbial cells and relevant metabolites within the environment such that the multistep process of lignocellulosic catabolism is disrupted. However, at lines 125-126, they state that they test this hypothesis through use of a pure *A. nidulans* culture. My question relates to how results from a pure culture can be used to test a hypothesis relating to metabolite transfer between (presumably) diverse organisms (many different species, not just a single species)? On lines 126-131, the authors even list a dichotomy between "previous studies' focus on microbial single species" and their focus on "the microbial community", but I don't see how investigation of a pure culture of *A. nidulans* is any different than the previous studies.
- 2) Overall, I find the descriptions of the experiments very lacking, and this must be corrected prior to manuscript publication. Instructions to the authors for this journal lists that "The Materials and Methods section should include sufficient technical information to allow the experiments to be repeated." I find that the methods described in this work could not be repeated by a scientist skilled in the art. For example, Just within the "Straw degradation under simulated microgravity and normal gravity section I came up with the following questions needing elaboration, if I were to try and repeat the experiments described: What was the moisture content of the straw prior to grinding? What was the particle size distribution following grinding? What type of grinder and at what settings and for how long was the material ground? How much material was ground at a time? Did dewatering the solid waste impact the concentration of viable microbial cells present in this waste material? What are the details for how the water, straw, and solid waste were mixed? If the petri dishes were not sealed, was there any loss of material from the cultures over the course of the experiment? How large were the samples taken over the course of the 25 day experiment? How were the samples taken (centrifuged prior to -80C storage, etc.)? The authors should address these details both for this section as well as for all the materials and methods sections in their manuscript.
- 3) Under Data Availability, the authors state "If reasonably requested, Dr. Jiajie Feng can be contacted to obtain the original materials." I do not find this acceptable. All raw sequencing data and metabolomics data should be deposited and freely available to readers using databases such as the NCBI Sequence Read Archive (SRA) and the UCSD Metabolomics Workbench, respectively. ASM open data policy and a list of potential public repositories is listed at <https://journals.asm.org/open-data-policy>
- 4) I applaud the authors for putting together a manuscript that is in large part clear to read. However, that said, there are points of the manuscript where grammatical mistakes are made which may confuse the reader. In my review, I have noted a few instances where clarity could be improved and offered some of my own suggestions, but I have not listed all instances and so I wanted to suggest that the authors could possibly benefit from using a service to proofread their manuscript prior to publication.

Specific Comments:

- Line 20: Perhaps "The weight loss of the straw and [breakdown] of its lignocellulosic contents were significantly slowed down"
- Line 39 Reword to "methods for more globalized microbial interactions" (deleting "a")
- Line 43: "Even for issues on the Earth" sounds clunky
- Line 52: Eliminate "The" to say simply state "Space exploration is an important cause"
- Line 75: Do the authors mean to state that O₂, CO₂, and H₂O cycles are relatively well-characterized, when they write "both of which have been relatively mature"? If so, they should rewrite this sentence for clarity.
- Lines 78-79: This sentence is also a little difficult to read. Perhaps rewrite to something like "Discarding organic solid waste directly to space would lead to pollution of the space environment (not conducive to planetary protection) and a considerable loss of potential life-support material"
- Line 112: The phrase "By the way" can be deleted here.
- Line 135: In the results section, the authors dive straight into the results of Figure 1 without introducing the rationale behind the experiment or providing general details relating to how the experiment was conducted. I would like the authors to try and introduce the question being asked and briefly summarizing how they are conducting the experiment designed to answer that question before summarizing the results from Figure 1. I have the same request for the "Metabolomics analysis" Section
- Line 138-139: If the culture are indeed becoming adapted to microgravity conditions, I would expect that if the cultures were taken the conditions tested and cells were reinoculated such as to mimic the experimental setup from Figure 1, that there would no longer be a discrepancy between microgravity and a 1G phenotypes. Did the authors attempt such an experiment? If the authors are going to suggest that their results after day 20, indicated a "likely" occurrence of adaptation, I would like the authors to expand on what experiments they performed to test this adaptation claim or at minimum suggest experiments that could test the claim and provide other hypotheses to explain their 20 day finding in the discussion.
- Line 149: As a personal preference, I would like the authors to migrate Panels A-C from Fig S3 to the main text as panels D-F of Figure 1, as I think these data should be presented alongside the data currently in Fig 1.
- Line 153: I don't like the use of the word "fiercely" here and while I agree that the pH phenotype is likely from ammonia release, can anything else other than ammonia cause the rise in pH that the authors saw such as consumption of organic acids present in the media, etc.?
- Lines 167-170: For which timepoints was the Shannon Index value calculated?
- Figure 3: I do not think it appropriate that the authors seemed to have taken the average log₂ fold change for Day 15 and Day 20, to report values listed in Figure 3. I would prefer that the authors keep these data separate and also provide error bars derived from replicate measurements. Additionally, it appears that Day 5 P_{CO2} separates the two conditions relatively well. I would like the authors to dive deeper into the data and provide some information relating to the loadings for P_{CO2} and discuss statistically significant metabolite differences observed at day 5 as part of their results section.
- Lines 198-199 and 206-207 and throughout the metabolomics section: I don't like the use of the term "down-regulated" in this section, since the data is derived from metabolomics data, it does not inform use directly on regulation of the metabolic

pathways, as a transcriptomics or proteomics experiment might, therefore, I might prefer "decreased in the microgravity samples" or similar and the authors should not make direct claims about regulation of involved pathways simply based upon metabolite abundance.

Line 237: I do not understand how growing *A. nidulans* on solid media helps inform the authors relating to the phenotypes observed for their mixed culture liquid media experiments. Can the authors expand on their rationale in this section?

Figure 5: While I understand the reasoning, I generally do not like the inclusion of faces for the microbes shown in panels A and B and I believe the same message can be conveyed without anthropomorphizing the graphic.

Reviewer #2 (Public repository details (Required)):

16S and ITS bacterial amplicon sequences.

Reviewer #2 (Comments for the Author):

The manuscript entitled "Simulated microgravity confines and fragments the straw-based lignocellulose degrading microbial community" by Liao, Feng et. al details work done to characterize the effect of microgravity on the lignocellulosic degradation capabilities of bacterial/fungal communities. In it, they use experimental systems to simulate microgravity, and then run a suite of analyses such as amplicon sequencing, metabolomics, physical and chemical testing, and enzymatic analyses to quantify the differences between the same community in different gravity environments. They also conduct longitudinal testing, which provides a rich set of data by which to identify differences in the trajectory of metabolic output and resource use. It is my opinion that the main findings of the text make sense and are supported by the data. I believe that this data set will be a great benefit to the scientific community and is related to an exciting and emerging area of microbiome science.

I find their proposed model for synthesizing the multiple individual results they observe to be very interesting and plausible, though perhaps requiring more analyses to fully support. I would like to see a few sentences added to the discussion which explore problems or shortcomings of the work and exploring potential alternate explanations. What other changes in microbial metabolism or physical complications could be driving this decreased rate in microgravity? What were some shortcomings to the experimental approach that was taken that might not fully model the microgravity environment? Is this truly a good model for a space-like environment without the addition of radiation, weakened magnetic field, and high vacuum? How could future experiments build on your work to correct for these shortcomings? I think an addition on this to the discussion would greatly increase the impact of the manuscript.

Futhermore, there are multiple places in the introduction that require significantly more primary citations to support the foundation for the claims made in this paper, pulling from both the smaller field of space-environment microbiology and from the much larger body of lignocellulosic degradation work using microbial communities.

Line Comments

Minor:

Lines 1-50: There are many statements in the first 5 sentences that require references, such as the SLS launch cost, etc. It would be useful to have at least one reference on the cost and difficulty of equipping crewed space travel.

76: pick number or amount, you do not need both.

87-88: This needs a reference.

92:94: This statement needs a reference.

110:112: This states that there are "numerous studies" on lignocellulosic degradation by microbial communities, but the authors cite no studies to support this claim. This claim is true but needs to be supported.

112:114: This sentence refers to "studies" (plural) but only references 1 study. This needs more studies to confirm the conflicting claim.

138: You say that after 20 days the rates of the two communities became more similar, and suggest that it is likely due to adaption. Is adaption the likely culprit? Could it be more likely to be the normal gravity group had simply expended most of the easy to metabolize resources? Also, I believe the best way to test this would be to take inoculants from the output of both of these experiments, and then test them again to more rigorously show that the microgravity condition has adapted to the environment by showing a similar rate to the 1G environment at earlier timepoints.

152-155: Is there a significance test you can use to test for the significance of this fold change?

165: I am confused, in the previous sentence you say there were ~2,100 bacterial ASVs found in the study, but in this sentence, you report ~200 ASVs, and a similar trend for fungi. Please clarify what is being reported in the text.

331:333: You suggest that mass-transfer-promoting methods should be adopted for globalization of microbial interactions required for lignocellulose degradation efficiency in space. Can you suggest a few options here?

345: Can you please not use the abbreviations, but explicitly write out the full names of the methods you reference.

Major:

FigureS5C is referenced in the text, saying that the fungal Shannon diversity in mu-g is lower than in 1G. The authors use

ANOVA to test for significance, but this does not test for a difference in the mean between two groups, and should not be used to try and determine if the average value of one group is lower than the other; all it shows is that the mu-g group has more variance in its values than the 1g group. This is an interesting finding, but the writing in the main text (167-169) is misleading and inaccurate. A more appropriate test would be a student's t-test or Wilcoxon. Given the lack of much significant difference in the fungal richness at most timepoints, it is very unlikely there is a true decrease in the average of the fungal Shannon index. Furthermore, the two groups overlap in their 25-75 interquartile range, further suggesting there is no real difference in the mean values of the two samples. There is also very little visual difference in the relative abundances of the bacterial phyla in the stacked bar plots between the two groups, with the exception of some of the earlier samples.

191:193: The authors state that the day 15 and 20 groups are the "most distinguished" from each other, but do not state how they determine this. I am assuming visually, but this is not a very quantitative or rigorous way of determining the difference between two groups via PCOA, they should perform PERMANOVA / ANOVA and test for a significant difference at all timepoints followed by Tukey's HSD pairwise comparisons, as it is entirely likely some of the other groups are just as significantly different from each other. At the very least, the significance test of the PERMANOVA test between the two groups at days 15 and 20 should be reported.

The manuscript entitled “Simulated microgravity confines and fragments the straw-based lignocellulose degrading microbial community” by Liao, Feng et. al details work done to characterize the effect of microgravity on the lignocellulosic degradation capabilities of bacterial/fungal communities. In it, they use experimental systems to simulate microgravity, and then run a suite of analyses such as amplicon sequencing, metabolomics, physical and chemical testing, and enzymatic analyses to quantify the differences between the same community in different gravity environments. They also conduct longitudinal testing, which provides a rich set of data by which to identify differences in the trajectory of metabolic output and resource use. It is my opinion that the main findings of the text make sense and are supported by the data. I believe that this data set will be a great benefit to the scientific community and is related to an exciting and emerging area of microbiome science.

I find their proposed model for synthesizing the multiple individual results they observe to be very interesting and plausible, though perhaps requiring more analyses to fully support. I would like to see a few sentences added to the discussion which explore problems or shortcomings of the work and exploring potential alternate explanations. What other changes in microbial metabolism or physical complications could be driving this decreased rate in microgravity? What were some shortcomings to the experimental approach that was taken that might not fully model the microgravity environment? Is this truly a good model for a space-like environment without the addition of radiation, weakened magnetic field, and high vacuum? How could future experiments build on your work to correct for these shortcomings? I think an addition on this to the discussion would greatly increase the impact of the manuscript.

Futhermore, there are multiple places in the introduction that require significantly more primary citations to support the foundation for the claims made in this paper, pulling from both the smaller field of space-environment microbiology and from the much larger body of lignocellulosic degradation work using microbial communities.

Line Comments

Minor:

Lines 1-50: There are many statements in the first 5 sentences that require references, such as the SLS launch cost, etc. It would be useful to have at least one reference on the cost and difficulty of equipping crewed space travel.

76: pick number or amount, you do not need both.

87-88: This needs a reference.

92:94: This statement needs a reference.

110:112: This states that there are “numerous studies” on lignocellulosic degradation by microbial communities, but the authors cite no studies to support this claim. This claim is true but needs to be supported.

112:114: This sentence refers to “studies” (plural) but only references 1 study. This needs more studies to confirm the conflicting claim.

138: You say that after 20 days the rates of the two communities became more similar, and suggest that it is likely due to adaption. Is adaption the likely culprit? Could it be

more likely to be the normal gravity group had simply expended most of the easy to metabolize resources? Also, I believe the best way to test this would be to take inoculants from the output of both of these experiments, and then test them again to more rigorously show that the microgravity condition has adapted to the environment by showing a similar rate to the 1G environment at earlier timepoints.

152-155: Is there a significance test you can use to test for the significance of this fold change?

165: I am confused, in the previous sentence you say there were ~2,100 bacterial ASVs found in the study, but in this sentence, you report ~200 ASVs, and a similar trend for fungi. Please clarify what is being reported in the text.

331:333: You suggest that mass-transfer-promoting methods should be adopted for globalization of microbial interactions required for lignocellulose degradation efficiency in space. Can you suggest a few options here?

345: Can you please not use the abbreviations, but explicitly write out the full names of the methods you reference.

Major:

FigureS5C is referenced in the text, saying that the fungal Shannon diversity in mu-g is lower than in 1G. The authors use ANOVA to test for significance, but this does not test for a difference in the mean between two groups, and should not be used to try and determine if the average value of one group is lower than the other; all it shows is that the mu-g group has more variance in its values than the 1g group. This is an interesting finding, but the writing in the main text (167-169) is misleading and inaccurate. A more appropriate test would be a student's t-test or Wilcoxon. Given the lack of much significant difference in the fungal richness at most timepoints, it is very unlikely there is a true decrease in the average of the fungal Shannon index. Furthermore, the two groups overlap in their 25-75 interquartile range, further suggesting there is no real difference in the mean values of the two samples. There is also very little visual difference in the relative abundances of the bacterial phyla in the stacked bar plots between the two groups, with the exception of some of the earlier samples.

191:193: The authors state that the day 15 and 20 groups are the "most distinguished" from each other, but do not state how they determine this. I am assuming visually, but this is not a very quantitative or rigorous way of determining the difference between two groups via PCOA, they should perform PERMANOVA / ANOVA and test for a significant difference at all timepoints followed by tukey's HSD pairwise comparisons, as it is entirely likely some of the other groups are just as significantly different from each other. At the very least, the significance test of the PERMANOVA test between the two groups at days 15 and 20 should be reported.

“Lunar Palace 1” team (Institute of Environmental Biology and Life Support Technology)
School of Biological Science and Medical Engineering, Beihang University
37 Xueyuan Road, Haidian District, Beijing 100191, China
Email: fengjiajie@buaa.edu.cn; *Web:* lss-lab.bme.buaa.edu.cn

To: Editor-in-Chief and Editor Han Jing of *Microbiology Spectrum*,

2024-12-18

Thank you very much for handling the review of our manuscript (Spectrum02466-24) entitled “Simulated microgravity confines and fragments the straw-based lignocellulose degrading microbial community” and your kind letter inviting us to resubmit a new, revised manuscript based on our previous version. We appreciate the insightful comments and suggestions of the two anonymous reviewers, and have carefully considered each point brought up by the reviewers. As a result, we have substantially revised the manuscript to address the comments. Our point-to-point replies are shown in the next pages.

Yours sincerely,

FENG Jiajie

Jiajie FENG, Ph.D., Associate Professor
School of Biological Science and Medical Engineering,
Beihang University, China

Reviewer #1 (Public repository details (Required)):

Sequencing and Metabolomics data both need to be included in a public repository

Response: It is our failure not depositing the available data. Now we have tried to deposit them, and succeeded for the sequencing data. But very unluckily, our metabolomics' mass spectrum mzML file was cleaned up from our bmKCloud and cannot be retrieved, which is required for public repositories such as UCSD Workbench. We have to provide the original data as a supplemental table which includes metabolite names, quantitative information, annotation information, m/z, retention time, mass error, etc. We believe this is sufficient for reproducing the results, and hope the reviewer could forgive our oversight. We have correspondingly rewritten the Data Availability as "Raw 16S rRNA gene and ITS amplicon sequences are available in NCBI SRA database (<http://www.ncbi.nlm.nih.gov/sra>) under study no. PRJNA1197767 and no. PRJNA1198645 respectively. Metabolomics data is available as Supplemental Table S2."

Reviewer #1 (Comments for the Author):

General Comments

1) One of the main conclusions from the authors seems to be that microgravity disrupts transport of diverse microbial cells and relevant metabolites within the environment such that the multistep process of lignocellulosic catabolism is disrupted. However, at lines 125-126, they state that they test this hypothesis through use of a pure *A. nidulans* culture. My question relates to how results from a pure culture can be used to test a hypothesis relating to metabolite transfer between (presumably) diverse organisms (many different species, not just a single species)? On lines 126-131, the authors even list a dichotomy between "previous studies' focus on microbial single species" and their focus on "the microbial community", but I don't see how investigation of a pure culture of *A. nidulans* is any different than the previous studies.

Response: We'd like to thank the anonymous reviewer for both encouragement and criticism! Additionally, we want to specially thank the reviewer for accepting the review, because we feel getting a manuscript reviewed is quite difficult nowadays. We agree with the reviewer that using a pure culture to "verify" the community's mechanism is farfetched. The pure culture experiment can only visualize or corroborate the microgravity's confinement on hyphae and colonies. Therefore, we have correspondingly changed to "We tried to partly visualize and explore the mechanism" in line 245. We have changed the pure-culture related sentences' wording from "verify/verification" to "visualize/visualization" or "corroborate" throughout the manuscript.

To avoid misunderstanding about the dichotomy the reviewer mentioned (at the end of the Intro), we have deleted the sentence "We then visually verified and explored this hypothesis by the pure culture of the lignocellulose degrader *Aspergillus nidulans*." at line 129.

2) Overall, I find the descriptions of the experiments very lacking, and this must be corrected prior to manuscript publication. Instructions to the authors for this journal lists that "The Materials and Methods section should include sufficient technical information to allow the experiments to be

repeated." I find that the methods described in this work could not be repeated by a scientist skilled in the art. For example, Just within the "Straw degradation under simulated microgravity and normal gravity section I came up with the following questions needing elaboration, if I were to try and repeat the experiments described: What was the moisture content of the straw prior to grinding? What was the particle size distribution following grinding? What type of grinder and at what settings and for how long was the material ground? How much material was ground at a time? Did dewatering the solid waste impact the concentration of viable microbial cells present in this waste material? What are the details for how the water, straw, and solid waste were mixed? If the petri dishes were not sealed, was there any loss of material from the cultures over the course of the experiment? How large were the samples taken over the course of the 25 day experiment? How were the samples taken (centrifuged prior to -80C storage, etc.)? The authors should address these details both for this section as well as for all the materials and methods sections in their manuscript.

Response: Many thanks to the reviewer for pointing out our oversight. We have followed the suggestion, and added the details for other researchers to reproduce our research by revising the corresponding paragraphs as:

“The fermentation substrate in this study was air-dried wheat straw whose the moisture content was <5%. The straw was ground by the FW100 grinder (Taisite, Tianjin, China) with 10 g of material added each time and ground for 30 s under the first gear. The ground material passing the 20-mesh sieve (particle size: 850 μm, Haiji Co. Ltd., Hengshui, Hebei, China) was collected for the experiment. The inoculum was the organic solid waste produced by our “Lunar Palace 365” experiment (3). Its major components were aerobic ferments of inedible plant parts (such as straw, old leaves and plant roots), human (crew) feces, and their accompanying microbes (41). To make the inoculum more representative, we mixed solid waste samples collected on July 4, 2017 (~the initial phase of the “Lunar Palace 365” experiment) and that on May 15, 2018 (~the final phase). Due to the high water content and large particles in the solid waste, to avoid insufficient contact with the crushed wheat straw, it was air dried for 48 h at 30°C until the water content reduced to less than 5%, and then ground to powder. Because air drying would be a typical treatment for straw and solid waste (3), the air-drying method in this study conforms to the application scenario’s effect on the material and its microbial community.

According to the mixing ratio with the highest degradation weight loss rate established by us earlier (8), water, straw and solid waste inoculum were mixed and fully stirred in a clear ziplock bag with a ratio of 65:35:7, and each 2.7 g was weighed and placed in a small petri dish (35 mm × 10 mm) as a biological replicate ($n = 3$). Petri dishes are without sealing film to ensure aerobic conditions and thus material exchange with outside, and deployed on the clinostats (Yulei Technology Co., LTD, Tianjin, China).”

in lines 408-428. We do not quite understand why the samples need to be centrifuged prior to -80C storage. We have addressed all the materials and methods sections in the manuscript.

3) Under Data Availability, the authors state "If reasonably requested, Dr. Jiajie Feng can be contacted to obtain the original materials." I do not find this acceptable. All raw sequencing data and metabolomics data should be deposited and freely available to readers using databases such as the NCBI Sequence Read Archive (SRA) and the UCSD Metabolomics Workbench, respectively.

ASM open data policy and a list of potential public repositories is listed at <https://journals.asm.org/open-data-policy>

Response: It is our failure not depositing the available data. Now we have tried to deposit them, and succeeded for the sequencing data. But very unluckily, our metabolomics' mass spectrum mzML file was cleaned up from our bmkCloud and cannot be retrieved, which is required for public repositories such as UCSD Workbench. We have to provide the original data as a supplemental table which includes metabolite names, quantitative information, annotation information, m/z, retention time, mass error, etc. We believe this is sufficient for reproducing the results, and hope the reviewer could forgive our oversight. We have correspondingly rewritten the Data Availability as "Raw 16S rRNA gene and ITS amplicon sequences are available in NCBI SRA database (<http://www.ncbi.nlm.nih.gov/sra>) under study no. PRJNA1197767 and no. PRJNA1198645 respectively. Metabolomics data is available as Supplemental Table S2."

4) I applaud the authors for putting together a manuscript that is in large part clear to read. However, that said, there are points of the manuscript where grammatical mistakes are made which may confuse the reader. In my review, I have noted a few instances where clarity could be improved and offered some of my own suggestions, but I have not listed all instances and so I wanted to suggest that the authors could possibly benefit from using a service to proofread their manuscript prior to publication.

Response: Many thanks to the reviewer's sincere suggestion on our grammar. We have thoroughly read the specific comments, generalized them, and correspondingly proofread the manuscript. We have also let a native-English-speaker colleague read the manuscript and make grammatical changes.

Specific Comments:

Line 20: Perhaps "The weight loss of the straw and [breakdown] of its lignocellulosic contents were significantly slowed down"

Response: According to Fig. 1, all weight losses were slowed down, including the straw itself, and its lignocellulosic contents. So we tend to think the original sentence as right. But to prevent potential misunderstandings, we have thereby revise the sentence as "The straw and its lignocellulose contents' weight losses were significantly slowed down by $\mu\text{-g}$ " in the main text.

Line 39 Reword to "methods for more globalized microbial interactions" (deleting "a")

Response: We have deleted "a" following the reviewer's comment.

Line 43: "Even for issues on the Earth" sounds clunky

Response: We have thereby revised the sentence as "For non-space-exploration scenarios".

Line 52: Eliminate "The" to say simply state "Space exploration is an important cause"

Response: We have revised as required. A wrong article indeed.

Line 75: Do the authors mean to state that O₂, CO₂, and H₂O cycles are relatively well-characterized, when they write "both of which have been relatively mature"? If so, they should rewrite this sentence for clarity.

Response: We considered again about this sentence in the manuscript and found itself not making sense, because gaseous and liquid wastes are difficult due to other impurities rather than O₂, CO₂ and H₂O. So we have deleted this statement. Other parts of the sentence are remained.

Lines 78-79: This sentence is also a little difficult to read. Perhaps rewrite to something like "Discarding organic solid waste directly to space would lead to pollution of the space environment (not conducive to planetary protection) and a considerable loss of potential life-support material"

Response: We have correspondingly revised as “Discarding it directly to space would lead to pollution to the space environment (not conducive to planetary protection) and a considerable loss of potential life-support materials”.

Line 112: The phrase "By the way" can be deleted here.

Response: We have deleted it as required.

Line 135: In the results section, the authors dive straight into the results of Figure 1 without introducing the rationale behind the experiment or providing general details relating to how the experiment was conducted. I would like the authors to try and introduce the question being asked and briefly summarizing how they are conducting the experiment designed to answer that question before summarizing the results from Figure 1. I have the same request for the "Metabolomics analysis" Section

Response: We agree, and we have correspondingly added “The fermentation material’s compositional degradation rates and possible related physical/chemical properties were measured.” to lines 140-141 (at the Fermentation material’s physical and chemical properties Section), and “To deeply illustrate the underlying mechanisms, the material’s metabolites and potential pathways were measured and analyzed.” to lines 197-198 (at the Metabolomics analysis Section).

Line 138-139: If the culture are indeed becoming adapted to microgravity conditions, I would expect that if the cultures were taken the conditions tested and cells were reinoculated such as to mimic the experimental setup from Figure 1, that there would no longer be a discrepancy between microgravity and a 1G phenotypes. Did the authors attempt such an experiment? If the authors are going to suggest that their results after day 20, indicated a "likely" occurrence of adaptation, I would like the authors to expand on what experiments they performed to test this adaptation claim or at minimum suggest experiments that could test the claim and provide other hypotheses to explain their 20 day finding in the discussion.

Response: We have deleted the mentioned statement “likely suggesting an adaptation to m μ -g” since no solid evidence could be found.

Line 149: As a personal preference, I would like the authors to migrate Panels A-C from Fig S3 to the main text as panels D-F of Figure 1, as I think these data should be presented alongside the data currently in Fig 1.

Response: We agree. It is indeed strange separating them. We have correspondingly moved Fig S3’s a-c to Fig 1’s d-f, moved the corresponding fig captions to lines 589-590, modified the corresponding and subsequent supp Figs’ serial numbers in the supp file and the main text.

Line 153: I don't like the use of the word "fiercely" here and while I agree that the pH phenotype is likely from ammonia release, can anything else other than ammonia cause the rise in pH that the

authors saw such as consumption of organic acids present in the media, etc.?

Response: We have changed the word “fiercely” to “substantially”. About the reason the pH rised, we agree with the reviewer and have added “or/and consumption of organic acids” to the line 161.

Lines 167-170: For which timepoints was the Shannon Index value calculated?

Response: It was all timepoints pooled together. To clarify this, we have added “All time points together” to this place (line 173).

Figure 3: I do not think it appropriate that the authors seemed to have taken the average log2 fold change for Day 15 and Day 20, to report values listed in Figure 3. I would prefer that the authors keep these data separate and also provide error bars derived from replicate measurements. Additionally, it appears that Day 5 PCo2 separates the two conditions relatively well. I would like the authors to dive deeper into the data and provide some information relating to the loadings for PCo2 and discuss statistically significant metabolite differences observed at day 5 as part of their results section.

Response: Actually the result in Fig 3 is not the average of day 15 and day 20, but the results of the two time points pooled together. To clarify this, we have changed the figure description from “results merged” to “results pooled together with the time points labeled” in line 600. To separate them (following the comment), we labeled “(day 15)” and “(day 20)” on Fig 3 (in both a and b subfigs) to tell the timepoints apart. The figure cannot have error bars because it is a log2-fold-change bar plot, which does not have traditional-way error bars (bars appearing in the fig already indicate significant differences), and this kind of figure seems widely-used in metabolomics.

We are sorry that we did not provide convincing reason choosing days 15 and 20 for finer observations instead of other time points. The authors lacked communication when writing this, and actually we have other supporting reasons on it. We agree that the Day 5’s PCo2 separates the groups well, but we found its corresponding metabolomics data not making sense and hard to explain. We have correspondingly revised the sentences as “The PCoA of the non-targeted metabolomics result showed that 1g and μ -g are distinguished on days 5, 15 and 20 of the fermentation (FIG. S6), and we found the metabolites of days 5, 10 and 25 lacking annotation information in mainstream databases such as KEGG or hardly hitting microbial-related pathways (Table S2), so we performed finer observations on days 15 and 20.” in lines 198-203.

We believe that it is more rigorous to use well-studied metabolites with well-annotated information for mapping, which makes it easier to reach consensus with scholars in the field and reduces unnecessary disagreements. Additionally, we speculate that Day 5 was around the start of the experiment and might be affected by random factors, and thus not representative for analyzing the major mechanisms.

Lines 198-199 and 206-207 and throughout the metabolomics section: I don't like the use of the term "down-regulated" in this section, since the data is derived from metabolomics data, it does not inform use directly on regulation of the metabolic pathways, as a transcriptomics or proteomics experiment might, therefore, I might prefer "decreased in the microgravity samples" or similar and the authors should not make direct claims about regulation of involved pathways simply based upon metabolite abundance.

Response: Thanks to the reviewer's comment which makes us realize the inappropriate wording. We have changed the "~regulated" word to words such as "enrich", "elevate" or "reduce" throughout the manuscript (including the legend in Fig. 3).

Line 237: I do not understand how growing A nidulans on solid media helps inform the authors relating to the phenotypes observed for their mixed culture liquid media experiments. Can the authors expand on their rationale in this section?

Response: Actually the major part of the experiment (using the straw) was not a liquid-media one, but also a mainly solid (because it is straw) and a little bit wet one, just like the solid-media A. nidulans experiment. We need to admit that the solid-media experiment only visualized a few aspects of the major experiment, such as the hyphae and the colony's extension. To clarify this, we have added "As the straw experiment was also done on the wet but solid straw material, we suppose such a solid-plate-media pure-culture experiment could be comparable and simplifying." to lines 249-251.

Figure 5: While I understand the reasoning, I generally do not like the inclusion of faces for the microbes shown in panels A and B and I believe the same message can be conveyed without anthropomorphizing the graphic.

Response: We have correspondingly removed the faces in Figure 5.

Reviewer #2 (Public repository details (Required)):

16S and ITS bacterial amplicon sequences.

Response: It is our failure not depositing the available data. We have deposited them appropriately and rewritten the Data Availability part as "Raw 16S rRNA gene and ITS amplicon sequences are available in NCBI SRA database (<http://www.ncbi.nlm.nih.gov/sra>) under study no. PRJNA1197767 and no. PRJNA1198645 respectively. Metabolomics data is available as Supplemental Table S2."

Reviewer #2 (Comments for the Author):

The manuscript entitled "Simulated microgravity confines and fragments the straw-based lignocellulose degrading microbial community" by Liao, Feng et. al details work done to characterize the effect of microgravity on the lignocellulosic degradation capabilities of bacterial/fungal communities. In it, they use experimental systems to simulate microgravity, and then run a suite of analyses such as amplicon sequencing, metabolomics, physical and chemical testing, and enzymatic analyses to quantify the differences between the same community in different gravity environments. They also conduct longitudinal testing, which provides a rich set of data by which to identify differences in the trajectory of metabolic output and resource use. It is

my opinion that the main findings of the text make sense and are supported by the data. I believe that this data set will be a great benefit to the scientific community and is related to an exciting and emerging area of microbiome science.

Response: Many thanks to the reviewer's kind comment! We will make persistent efforts on this area. Additionally, we want to specially thank the reviewer for accepting the review. We feel getting a manuscript reviewed is quite difficult nowadays.

I find their proposed model for synthesizing the multiple individual results they observe to be very interesting and plausible, though perhaps requiring more analyses to fully support. I would like to see a few sentences added to the discussion which explore problems or shortcomings of the work and exploring potential alternate explanations. What other changes in microbial metabolism or physical complications could be driving this decreased rate in microgravity? What were some shortcomings to the experimental approach that was taken that might not fully model the microgravity environment? Is this truly a good model for a space-like environment without the addition of radiation, weakened magnetic field, and high vacuum? How could future experiments build on your work to correct for these shortcomings? I think an addition on this to the discussion would greatly increase the impact of the manuscript.

Response: Thanks for the inspiring comment! About the microgravity simulator (and space-like environment) issue, based on our investigation, it is a large, thoroughly discussed topic and maybe not proper to expand here. To address this, we tend to guide the readers to read existing reports, by adding a sentence "This study simulated microgravity by clinostats, whose results have been shown as similar to real microgravity and also thoroughly compared to other simulators on the ground (Huang, Li et al. 2018)." at the top of Discussion (line 276-278).

About interconnecting with other (reported/potential) metabolic or physical mechanisms and corresponding future experiments, we investigated deeper on existing studies, but did find the current study's complicated and unique scenario leads to the difficulty interconnecting with previous studies. But we did get some more inspirations. We have added a paragraph "Studies on microbes under microgravity have been abundant, but showed plausible conflicting and diverse results, likely due to the "indirect" manner by which μ -g/m μ -g affect microbes, via intermediate factors such as microbial species/strain types, cell motility, culture methods (suspension or agar cultures), nutrition availability, and culture time (Huang, Li et al. 2018). Therefore, the current study's complicated and unique scenario leads to the difficulty interconnecting with previous studies. An interesting "pure-culture network" experiment (Kato, Haruta et al. 2008) may inspire the follow-up mechanistic study of the current one: a lignocellulose-based pure-culture of the representative species under m μ -g, including each step's degraders and antimicrobial metabolites producers, could verify our conclusions and explore deeper." to lines 381-389 (Discussion).

Futhermore, there are multiple places in the introduction that require significantly more primary citations to support the foundation for the claims made in this paper, pulling from both the smaller field of space-environment microbiology and from the much larger body of lignocellulosic degradation work using microbial communities.

Response: We have correspondingly added the citations according to the reviewer's comments.

Line Comments

Minor:

Lines 1-50: There are many statements in the first 5 sentences that require references, such as the SLS launch cost, etc. It would be useful to have at least one reference on the cost and difficulty of equipping crewed space travel.

Response: We have added the reference “GAO US. Space Launch System: Cost Transparency Needed to Monitor Program Affordability. Washington, D.C.; 2023. No.: GAO-23-105609” following the reviewer’s suggestion. And changed the sentence as “For example, the United States’ Space Launch System (SLS) is able to transport 70–130 tons of materials into Earth-Moon transfer orbit each time, but the launch cost could be ~\$11.8 billion (GAO 2023).”

76: pick number or amount, you do not need both.

Response: We have followed the comment and deleted the word “number”.

87-88: This needs a reference.

Response: We have added “Liu H, Yao Z, Fu Y, Feng J. 2021. Review of research into Bioregenerative Life Support System(s) which can support humans living in space. Life Sciences in Space Research 31. <https://doi.org/10.1016/j.lssr.2021.09.003>” as the reference here, in line 94.

92:94: This statement needs a reference.

Response: We have added “Cui J, Yi Z, Chen D, Fu Y, Liu H. 2023. Microgravity stress alters bacterial community assembly and co-occurrence networks during wheat seed germination. Science of The Total Environment 890:164147.” as the reference here, in line 99.

110:112: This states that there are "numerous studies" on lignocellulosic degradation by microbial communities, but the authors cite no studies to support this claim. This claim is true but needs to be supported.

Response: We have added “Huang B, Li D-G, Huang Y, Liu C-T. 2018. Effects of spaceflight and simulated microgravity on microbial growth and secondary metabolism. Military Medical Research 5(1):18. <https://doi.org/10.1186/s40779-018-0162-9>” as the reference here, in line 116. The cited article is a review, which claimed there are “abundant studies”. After all, we think citing many studies here would introduce too many extra references.

112:114: This sentence refers to "studies" (plural) but only references 1 study. This needs more studies to confirm the conflicting claim.

Response: We considered again and think the current sentence does not need to appear here, so we have deleted the original whole sentence (at line 117).

138: You say that after 20 days the rates of the two communities became more similar, and suggest that it is likely due to adaption. Is adaption the likely culprit? Could it be more likely to be the normal gravity group had simply expended most of the easy to metabolize resources? Also, I believe the best way to test this would be to take inoculants from the output of both of these experiments, and then test them again to more rigorously show that the microgravity condition has adapted to the environment by showing a similar rate to the 1G environment at earlier timepoints.

Response: We have deleted the mentioned statement “likely suggesting an adaptation to $\mu\text{-g}$ ” since no solid evidence could be found.

152-155: Is there a significance test you can use to test for the significance of this fold change?

Response: We have analyzed with paired student’s *t*-test and added “ $P<0.001$ ” to the line 159.

165: I am confused, in the previous sentence you say there were ~2,100 bacterial ASVs found in the study, but in this sentence, you report ~200 ASVs, and a similar trend for fungi. Please clarify what is being reported in the text.

Response: The ~2,100 ASVs are the total number across all samples. To clarify this, we have added “per sample” to lines 171-172 as “~400 ASVs per sample” and “<200 ASVs per sample”.

331:333: You suggest that mass-transfer-promoting methods should be adopted for globalization of microbial interactions required for lignocellulose degradation efficiency in space. Can you suggest a few options here?

Response: We have added “such as stirring, heat gradient, moisture gradient, etc.” to the line 345-346.

345: Can you please not use the abbreviations, but explicitly write out the full names of the methods you reference.

Response: We have correspondingly changed “RWBs” to “rotating wall bioreactors”, and “HARVs” to “high-aspect rotating vessels” in line 358-359.

Major:

FigureS5C is referenced in the text, saying that the fungal Shannon diversity in $\mu\text{-g}$ is lower than in 1G. The authors use ANOVA to test for significance, but this does not test for a difference in the mean between two groups, and should not be used to try and determine if the average value of one group is lower than the other; all it shows is that the $\mu\text{-g}$ group has more variance in its values than the 1g group. This is an interesting finding, but the writing in the main text (167-169) is misleading and inaccurate. A more appropriate test would be a student's *t*-test or Wilcoxon. Given the lack of much significant difference in the fungal richness at most timepoints, it is very unlikely there is a true decrease in the average of the fungal Shannon index. Furthermore, the two groups overlap in their 25-75 interquartile range, further suggesting there is no real difference in the mean values of the two samples. There is also very little visual difference in the relative abundances of the bacterial phyla in the stacked bar plots between the two groups, with the exception of some of the earlier samples.

Response: Thanks to the correction! We have redone the analysis in paired student’s *t*-test and found the P value as 0.0487. We may have meant “one-way ANOVA” in this place when writing the statistical analysis part. To clarify this, we have added “paired student’s *t*-test” into the brackets in line 175, as “Shannon diversity index of fungi significantly decreased under $\mu\text{-g}$ ($P<0.05$, paired student’s *t*-test, FIG. S4c)”, and correspondingly revised the fig legends of Fig. S4c.

191:193: The authors state that the day 15 and 20 groups are the "most distinguished" from each

other, but do not state how they determine this. I am assuming visually, but this is not a very quantitative or rigorous way of determining the difference between two groups via PCOA, they should perform PERMANOVA / ANOVA and test for a significant difference at all timepoints followed by tukey's HSD pairwise comparisons, as it is entirely likely some of the other groups are just as significantly different from each other. At the very least, the significance test of the PERMANOVA test between the two groups at days 15 and 20 should be reported.

Response: We are sorry that we did not provide convincing reason choosing days 15 and 20 for finer observations instead of other time points. The authors lacked communication when writing this, and actually we have other supporting reasons on it. And we did not state that day 15 and 20 are “most distinguished from each other”, instead we stated their 1g and m μ -g are the most distinguished on these two days (in line 199).

We have correspondingly revised the sentences as “The PCoA of the non-targeted metabolomics result showed that 1g and m μ -g are distinguished on days 5, 15 and 20 of the fermentation (FIG. S6), and we found the metabolites of days 5, 10 and 25 lacking annotation information in mainstream databases such as KEGG or hardly hitting microbial-related pathways (Table S2), so we performed finer observations on days 15 and 20.” in lines 198-203.

We believe that it is more rigorous to use well-studied metabolites with well-annotated information for mapping, which makes it easier to reach consensus with scholars in the field and reduces unnecessary disagreements. Additionally, we suspect the day 5 is too close to the starting point and have some random effects not worth discussing.

Re: Spectrum02466-24R1 (Simulated microgravity confines and fragments the straw-based lignocellulose degrading microbial community)

Dear Dr. Jiajie Feng:

Thank you for the privilege of reviewing your work. Below you will find my comments, instructions from the Spectrum editorial office, and the reviewer comments.

Revision Guidelines

Sincerely,
Jing Han
Editor
Microbiology Spectrum

Reviewer #1 (Comments for the Author):

I thank the authors for taking time to address many of the comments from myself and the other reviewer of this work. In large part, the authors have improved the readability and the content of their manuscript. I also thank the authors for making their raw data publicly available, which is critical for publication. That said, I still have a few lingering comments which I believe should be addressed prior to publication of this work:

Figure 3: I will refer to my previous comment regarding separating the Day 15 and Day 20 data. That is, for Figure 3a, I would like to see the log₂ fold-change values for all 20 metabolites mentioned for both Day 15 and Day 20. For instance, I would like to see the log₂ fold change for cyclohexylamine both for Day 15 and for Day 20. Additionally, the authors claimed in their response that a log₂ fold-change plot does not have traditional error bars. I dispute this. The authors claim that 3 biological replicates were used during the experiment. Presumably (but the authors need to make this clearer in their work) the fold-change values represent a fold-change for the listed metabolites between day 0 (inoculation) and either day 15 or day 20 OR between mu-g and 1g at day 15 or day 20 (both of which have three biological replicates, and I assume were inoculated using the same inoculum). In either case, comparing replicate 1 for the test condition vs replicate 1 for the control condition will provide a fold-change and iterating this through this process with each of three replicates will provide three different fold changes which can be averaged (which I assume is the value listed in Figure 3). In this case the calculated average fold-change will also have a standard deviation associated with it and this should be provided as error bars on the figure.

Materials and Methods: While I thank the authors for making changes to the "Straw degradation under simulated microgravity" Methods section, There are still some areas that seem to lack adequate explanations for the remained of the materials and methods sections. For example, what size samples were taken for DNA extraction? How large of samples were freeze dried for metabolomics experiments and what are the details of this freeze drying process? What is "an appropriate" amount of methanol? What are the conditions for centrifugation and what type and size of filter was used for the metabolomics samples? What were the liquid chromatography run conditions? What was the mobile phase used for LC analysis? What are the important settings for the mass spectrometer? I want to reiterate that the authors need to provide sufficient details to repeat all experiments within the materials and methods section. If I wanted to conduct the same experiment myself, I should have all the information needed to do this based on the text within the materials and methods section. Currently, this is not the case.

Lines 162-163: I do not think that "Given the common sense of fungi's dominant role in lignocellulose degradation" is an appropriate statement here. The others should say something more like "given fungi's well-characterized central role in lignocellulose degradation" but must also include citations for this statement

Line 163-165: This sentence is very clunky as written which confuses this reviewer on what the authors mean to say.

Line 165: The "S" is Student's t-test must be capitalized

Line 174: What are topological properties of "small world" and "scale free"?

Line 183: I do not like the inclusion of statements such as "it is obvious" please eliminate. Perhaps replace with "the data indicate"

“Lunar Palace 1” team (Institute of Environmental Biology and Life Support Technology)
School of Biological Science and Medical Engineering, Beihang University
37 Xueyuan Road, Haidian District, Beijing 100191, China
Email: fengjiajie@buaa.edu.cn; *Web:* lss-lab.bme.buaa.edu.cn

To: Editor-in-Chief and Editor Han Jing of *Microbiology Spectrum*,

2025-2-23

Thank you very much for handling the review of our manuscript (Spectrum02466-24R1) entitled “Simulated microgravity confines and fragments the straw-based lignocellulose degrading microbial community” and your kind letter inviting us to resubmit a new, revised manuscript (3rd) based on our 2nd version. We greatly appreciate the valuable feedback provided by Reviewer 1 and have made corresponding revisions to the manuscript based on the previous round of modifications. Our point-to-point replies are shown in the response letter.

Yours sincerely,

FENG Jiajie

Jiajie FENG, Ph.D., Associate Professor

School of Biological Science and Medical Engineering,

Beihang University, China

Reviewer #1:

Comment 1: Figure 3: I will refer to my previous comment regarding separating the Day 15 and Day 20 data. That is, for Figure 3a, I would like to see the log₂ fold-change values for all 20 metabolites mentioned for both Day 15 and Day 20. For instance, I would like to see the log₂ fold change for cyclohexylamine both for Day 15 and for Day 20. Additionally, the authors claimed in their response that a log₂ fold-change plot does not have traditional error bars. I dispute this. The authors claim that 3 biological replicates were used during the experiment. Presumably (but the authors need to make this clearer in their work) the fold-change values represent a fold-change for the listed metabolites between day 0 (inoculation) and either day 15 or day 20 OR between m μ -g and 1g at day 15 or day 20 (both of which have three biological replicates, and I assume were inoculated using the same inoculum). In either case, comparing replicate 1 for the test condition vs replicate 1 for the control condition will provide a fold-change and iterating this through this process with each of three replicates will provide three different fold changes which can be averaged (which I assume is the value listed in Figure 3). In this case the calculated average fold-change will also have a standard deviation associated with it and this should be provided as error bars on the figure.

Response: Thank you very much for your valuable feedback. Based on your suggestions, we have updated Figure 3a to show the upregulation and downregulation of the 20 metabolites under m μ -g conditions at Day 15 and Day 20. We have also added error bars as per your recommended calculation method. The revised Figure 3a is shown below:

Accordingly, we have also revised the caption for Figure 3. In lines 634-636, we added: “error bars represent standard error ($n = 3$ biological replicates), numbers indicate the log₂(fold change) values for each metabolite.”

Comment 2: Materials and Methods: While I thank the authors for making changes to the "Straw degradation under simulated microgravity" Methods section, There are still some areas that seem to lack adequate explanations for the remained of the materials and methods sections. For example, what size samples were taken for DNA extraction? How large of samples were freeze dried for metabolomics experiments and what are the details of this freeze drying process? What is "an appropriate" amount of methanol? What are the conditions for centrifugation and what type and

size of filter was used for the metabolomics samples? What were the liquid chromatography run conditions? What was the mobile phase used for LC analysis? What are the important settings for the mass spectrometer? I want to reiterate that the authors need to provide sufficient details to repeat all experiments within the materials and methods section. If I wanted to conduct the same experiment myself, I should have all the information needed to do this based on the text within the materials and methods section. Currently, this is not the case.

Response: Thank you very much for your suggestions. We acknowledge that our description of the experimental procedures was insufficient. To ensure the reproducibility of the experiments, we have rewritten the section on “non-targeted metabolomics”. In lines 482-524, we have included the following detailed description:

“Non-targeted metabolite profiling of fermentation samples was conducted by Biomarker Technologies Corporation, Beijing, China. The fermentation samples (50 mg) were freeze-dried and ground into a fine powder. The powdered samples were then dissolved in 1000 μ L of extraction solvent (methanol : acetonitrile : water = 2 : 2 : 1, v/v/v) containing an internal standard (2-chloro-L-phenylalanine, 20 mg/L). The mixture was vortexed for 30 seconds, homogenized using a grinder at 45 Hz for 10 minutes, and sonicated in an ice-water bath for 10 minutes. After incubation at -20°C for 1 hour, the samples were centrifuged at 12,000 rpm for 15 minutes at 4°C. The supernatant (500 μ L) was collected, dried using a vacuum concentrator, and reconstituted in 160 μ L of reconstitution solvent (acetonitrile:water = 1:1, v/v). The final extract was centrifuged again, and 120 μ L of the supernatant was transferred to injection vials for LC-MS analysis. Each sample was combined with 10 μ L from other samples to create a quality control (QC) sample for machine analysis.

Metabolite analysis was performed using a Waters Acquity I-Class PLUS ultra-high-performance liquid chromatography (UPLC) system coupled with a Waters Xevo G2-XS QTOF high-resolution mass spectrometer. Chromatographic separation was achieved using a Waters Acquity UPLC HSS T3 column (1.8 μ m, 2.1 \times 100 mm). The mobile phase consisted of 0.1% formic acid in water (mobile phase A) and 0.1% formic acid in acetonitrile (mobile phase B) for both positive and negative ion modes. The gradient program was as follows: 0–0.25 min, 98% A; 0.25–10 min, linear gradient to 2% A; 10–13 min, 2% A; 13.1–15 min, return to 98% A. The flow rate was maintained at 400 μ L/min. Mass spectrometry data were acquired in MSe mode, with low collision energy set at 2 V and high collision energy ranging from 10 to 40 V. The scanning frequency was 0.2 seconds per spectrum. The ESI ion source parameters were as follows: capillary voltage at 2000 V (positive ion mode) or -1500 V (negative ion mode), cone voltage at 30 V, ion source temperature at 150°C, desolvation gas temperature at 500°C, backflush gas flow rate at 50 L/h, and desolvation gas flow rate at 800 L/h. The mass-to-charge ratio (m/z) range was set to 50–1200.

The raw data collected using MassLynx V4.2 software were processed using Progenesis QI software for peak extraction, peak alignment, and other data processing operations. Metabolite identification was performed using the METLIN (METabolite LINKage Database), KEGG (Kyoto Encyclopedia of Genes and Genomes), HMDB (Human Metabolome Database) and LIPID MAPS (Lipid Metabolites and Pathways Strategy) databases. Theoretical fragment identification was conducted with a mass deviation within 100 ppm for precursor ions and 50 ppm for fragment ions. To preliminarily understand the overall metabolic differences among each group of samples,

Orthogonal Projections to Latent Structures-Discriminant Analysis (OPLS-DA) were conducted on metabolome data using the R software package “ropls” with a 200-time permutation to verify the model’s reliability. A t-test was used to calculate significant differences between treatment groups for each metabolite. The VIP value of the model was calculated using multiple cross-validation. Differential metabolites were screened based on the criteria of $|\log_2(\text{fold change})| > 0.9$, $P < 0.05$, and $VIP > 1$. The differential metabolites of KEGG pathway enrichment significance were calculated using a hypergeometric distribution test (R package “clusterProfiler”).”

For the DNA extraction procedure, we have added the following details in lines 451-454 of the document: “Briefly, Total genomic DNA was extracted from 0.5 g fermentation samples using the TGuide S96 Magnetic Soil / Stool DNA Kit (Tiangen Biotech (Beijing) Co., Ltd., China) according to the manufacturer’s instruction.”

Additionally, for the *Aspergillus nidulans* plate inoculation procedure, we have added the following details in lines 573-577: “50 mg of freeze-dried powder of the strain was completely dissolved in 5 mL PBS buffer. The dissolved fungal solution was then diluted into a series of concentrations ranging from 10^{-1} to 10^{-7} , and spread onto on sterile carboxymethyl cellulose solid medium plates with 100 μL each”. in line 583-584, we added: “Plates with a dilution factor of 10^{-4} , where the colony edges were clear and the number of colonies was moderate, were selected for photography”.

Comment 3: Lines 162-163: I do not think that "Given the common sense of fungi’s dominant role in lignocellulose degradation" is an appropriate statement here. The others should say something more like "given fungi’s well-characterized central role in lignocellulose degradation" but must also include citations for this statement.

Response: Thank you for your reminder. We have revised the statement accordingly. In lines 159-162 of the document, we have rewritten it as: “In our study, fungi (~400 ASVs per sample, FIG. S4a) showed a much higher richness than bacteria (< 200 ASVs per sample, FIG. S4b) in both 1g and $\mu\text{-g}$ conditions, given fungi’s well-characterized central role in lignocellulose degradation (13)”. We have also added the corresponding reference in lines 684-685:

“13. Kirchman DL. 2018. Degradation of organic matter. Processes in Microbial Ecology, 2nd edn. Oxford University Press. <https://doi.org/10.1093/oso/9780198789406.003.0007>.”

Comment 4: Line 163-165: This sentence is very clunky as written which confuses this reviewer on what the authors mean to say.

Response: We apologize for the confusion caused by the previous wording. To improve readability, we have rewritten this section. In lines 166-174, we have revised it as: “Notably, the overall Shannon diversity index of fungi was significantly reduced under $\mu\text{-g}$ ($P < 0.05$, paired Student’s t-test, FIG. S4c), which may partially explain the suppressed degradation of straw under $\mu\text{-g}$ conditions (FIG. 1). In terms of microbial community composition, bacterial taxonomic annotation showed dominance by three phyla: *Actinobacteria*, *Firmicutes* and *Proteobacteria*, with *Proteobacteria* gradually increasing in relative abundance over time (from ~0% to ~80%, FIG. S5a). Meanwhile, fungal communities were consistently dominated by the phylum *Ascomycota* (FIG. S5b).”

Comment 5: Line 165: The "S" is Student's t-test must be capitalized

Response: Thank you for pointing this out. We have corrected the capitalization in line 167 of the document. "Student's *t*-test" now has the "S" capitalized.

Comment 6: Line 174: What are topological properties of "small world" and "scale free"?

Response: We apologize for the confusion. The terms "small world", "scale-free", and "modularity" refer to typical topological properties of networks. "Small world" indicates that the network has a short average path length and a high clustering coefficient. "Scale-free" means that the node degree follows a power-law distribution (i.e., a few nodes have many connections, while most nodes have few connections). "Modularity" refers to the presence of tightly connected subgroups (modules) within the network, where connections within modules are dense, and connections between modules are sparse. One advantage of the molecular ecological networks we constructed based on the RMT theory is that by comparing the statistical differences between the real network and the RMT random network, we can determine whether the constructed network exhibits these typical topological properties. To reduce reader confusion, we have added references 14 and 15 after the sentence in lines 178-179: "both networks showed topological properties of small world, scale-free, and modularity (14, 15)."

The citation details are provided in lines 686-689:

14. Barabási A-L, Oltvai ZN. 2004. Network biology: understanding the cell's functional organization. *Nature Reviews Genetics* 5(2):101-13. <https://doi.org/10.1038/nrg1272>

15. Deng Y, Jiang YH, Yang Y, He Z, Luo F, Zhou J. 2012. Molecular ecological network analyses. *BMC bioinformatics* 13:113. <https://doi.org/10.1186/1471-2105-13-113>

Comment 7: Line 183: I do not like the inclusion of statements such as "it is obvious" please eliminate. Perhaps replace with "the data indicate".

Response: Thank you for your suggestion. We have revised the statement in lines 186-187 to: "Consistently, the data indicate that".

Re: Spectrum02466-24R2 (Simulated microgravity confines and fragments the straw-based lignocellulose degrading microbial community)

Dear Dr. Jiajie Feng:

Your manuscript has been accepted, and I am forwarding it to the ASM production staff for publication. Your paper will first be checked to make sure all elements meet the technical requirements. ASM staff will contact you if anything needs to be revised before copyediting and production can begin. Otherwise, you will be notified when your proofs are ready to be viewed.

Sincerely,
Jing Han
Editor
Microbiology Spectrum